# MEASUREMENT-ALIGNED SAMPLING FOR INVERSE PROBLEMS

## ABSTRACT

Diffusion models provide a powerful way to incorporate complex prior information for solving inverse problems. However, existing methods struggle to correctly incorporate guidance from conflicting signals in the prior and measurement, and often failed to maximizing the consistency to the measurement, especially in the challenging setting of non-Gaussian or unknown noise. To address these issues, we propose Measurement-Aligned Sampling (MAS), a novel framework for linear inverse problem solving that flexibly balances prior and measurement information. MAS unifies and extends existing approaches such as DDNM, TMPD, while generalizing to handle both known Gaussian noise and unknown or non-Gaussian noise types. Extensive experiments demonstrate that MAS consistently outperforms state-of-the-art methods across a variety of tasks, while maintaining relatively low computational cost.

## 1 INTRODUCTION

Inverse problems are prevalent in image restoration (IR) tasks, including super-resolution, inpainting, deblurring, colorization, denoising, and JPEG restoration (Chung et al., 2022a; Kawar et al., 2022a; Saharia et al., 2022; Wang et al., 2022; Lugmayr et al., 2022; Mardani et al., 2023; Song et al., 2023b; Kawar et al., 2022b). Solving an inverse problem involves recovering an unknown original image $x_0 \in \mathbb{R}^n$ based on information from a prior distribution, $\pi(x_0)$, and noisy measurements $y \in \mathbb{R}^m$ generated through a forward model:

$$y = \mathcal{H}(x_0) + \epsilon. \tag{1}$$

Here $\epsilon \in \mathbb{R}^m$ represents measurement noise, $x_0 \in \mathbb{R}^d$ is drawn from data distribution $\pi_0(x_0)$, $\mathcal{H} : \mathbb{R}^d \mapsto \mathbb{R}^m$ is the measurement function, and $y \in \mathbb{R}^m$ denotes the degraded measurement or observed image. A useful motivating example is a high-resolution image $x_0$, with a noisy degraded image $y$ and a known corruption process.

Pretrained diffusion and flow models offer a prior distribution $\pi_0(x_0)$ that greatly aids in solving inverse problems. Methods such as DPS (Chung et al., 2022a), ΠGDM (Song et al., 2023b), and TMPD (Boys et al., 2023) estimate conditional scores directly from the measurement model by leveraging score decomposition to guide each diffusion sampling step. In contrast, approaches like FPS (Dou & Song, 2024), DAPS (Zhang et al., 2024), MPGD (He et al., 2023), and optimization-based methods (Song et al., 2023a; Zhu et al., 2023; Li et al., 2024; Wang et al., 2024) align denoiser outputs directly with measurements, thereby avoiding backpropagation through the U-Net. Although DAPS achieves state-of-the-art performance—outperforming methods that require backpropagation—it still requires more than 100 gradient descent iterations per diffusion step, making it far more computationally expensive compared to methods such as DDNM (Wang et al., 2022) and DDRM (Kawar et al., 2022a). This highlights the promise of developing approaches that avoid both backpropagation through U-Net and excessive optimization steps, while still attaining state-of-the-art performance.

Moreover, the above approaches lack the ability to effectively handle unknown or non-Gaussian noise. In practical settings, noise frequently deviates from Gaussian assumptions—exhibiting characteristics like salt-and-pepper, periodic, or Poisson distributions—or is completely unknown. Additionally, the forward measurement operator may be uncertain or inaccurately specified. Effectively addressing

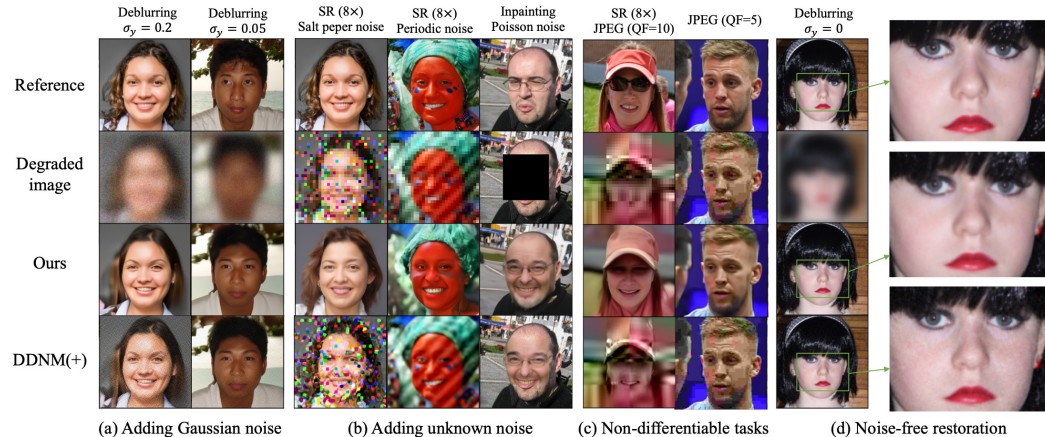

| | Deblurring $\sigma_y = 0.2$ | Deblurring $\sigma_y = 0.05$ | SR (8×) Salt peper noise | SR (8×) Periodic noise | Inpainting Poisson noise | SR (8×) JPEG (QF=10) | JPEG (QF=5) | Deblurring $\sigma_y = 0$ | |
|---|---|---|---|---|---|---|---|---|---|
| Reference | | | | | | | | | |
| Degraded image | | | | | | | | | |
| Ours | | | | | | | | | |
| DDNM(+) | | | | | | | | | |
| | (a) Adding Gaussian noise | | (b) Adding unknown noise | | | (c) Non-differentiable tasks | | (d) Noise-free restoration | |

Figure 1: Solving various inverse problems using unconditional diffusion models. Our model demonstrates better robustness with unknown noise and strong Gaussian noise.

Table 1: Comparison of method applicability across different inverse problems.

| Inverse Problem | Noise strength | DDNM | DDRM | ΠGDM | DAPS | RED-Diff | MAS (ours) |
|---|---|---|---|---|---|---|---|
| Linear + noise free | - | ✓ | ✓ | ✓ | ✓ | ✓ | ✓ |
| Linear + Gaussian noise | Known | ✓ | ✓ | ✓ | ✓ | ✓ | ✓ |
| Linear + non-Gaussian noise | Unknown | ✗ | ✗ | ✗ | ✗ | ✗ | ✓ |
| JPEG / Quantization restoration | Known | ✗ | ✗ | ✓ | ✗ | ✗ | ✓ |
| JPEG / Quantization restoration | Unknown | ✗ | ✗ | ✗ | ✗ | ✗ | ✓ |

inverse problems under these more general and realistic conditions remains an open and challenging research area.

Our main contributions are summarized as follows:

- We propose *Measurement-Aligned Sampling* (MAS), a novel framework for solving linear inverse problems. MAS provides both probabilistic and optimization perspectives and generalizes approaches such as DDNM and TMPD for linear inverse problems. Furthermore, our proposed 'overshooting' technique achieves superior restoration quality compared to DDNM across various inverse problem scenarios.

- We develop new techniques that *maximize consistency with the measurement*, enabling robust handling of both Gaussian noise and unknown noise sources. Moreover, our novel parameterization scheme allows us to effectively handle noisy inverse problems with unknown or non-Gaussian noise structures and even non-differentiable measurements, such as JPEG restoration, without requiring explicit knowledge of the forward operator or noise level. The comparison of method applicability across different inverse problems is shown in Table 1.

- Our experiments show that MAS enables robust and efficient image restoration, consistently outperforming baselines across Gaussian, non-Gaussian, and non-differentiable degradations (see Fig. 1 and experiments in Sec. 5).

## 2 BACKGROUND

Given training dataset $\mathcal{D} = \{x_0^i\}_{i=1}^N$ from target distribution $\pi_0(x_0)$, $x_0^i \in \mathbb{R}^d$, the goal of generative modeling is to draw new samples from $\pi_0$. In the context of conditional generation, suppose that we have data samples from a joint distribution $(x_0^i, y) \sim \pi(x_0, y)$, where $x_0, y$ are dependent, and $y$ could be class labels or text information, for example.

For conditional generative modeling, we seek to draw new samples from $\pi(x_0 \mid y)$ for a given condition $y$. Conditioned flows (Zheng et al., 2023) build a marginal probability path $p_{t|y}$ using a

mixture of interpolating densities: $p_{t|y}(x_t \mid y) = \int p_t(x_t \mid x_0)\pi(x_0 \mid y)dx_T$, where $p_t(\cdot \mid x_0)$ is a probability path interpolating between noise and a single data point $x_T$. In general, the conditional kernel $p_t(x_t \mid x_0)$ is given by a Gaussian distribution: $p_t(x_t \mid x_0) = \mathcal{N}(x_t; \alpha_t x_0, \sigma_t^2 \mathbb{I})$, where $\mathcal{N}$ is the Gaussian kernel, $\alpha_t, \sigma_t$ are differentiable functions. Then we can sample from the conditional distribution $p_{0|y}(x_0 \mid y)$ by simulating a stochastic process $p_{t|y}(x_t \mid y)$ from time $t = T$ to $t = 0$. Although different sampling methods can be chosen, generally, the iteration follows the form:

$$x_{t-\Delta t} \sim \mathcal{N}(a_t m_{0|t,y} + b_t x_t, c_t^2 \mathbb{I}). \tag{2}$$

where $m_{0|t,y} = \mathbb{E}[x_0 \mid x_t, y]$ is the idea conditional denoiser, $a_t$, $b_t$ and $c_t$ are parameters that depends on samplers. For instance, $x_{t-\Delta t} \sim \mathcal{N}(\alpha_{t-\Delta t} m_{0|t,y}, \sigma_{t-\Delta t} \mathbb{I})$ is a valid DDIM sampler. In the implementation of conditional diffusion models, a denoiser is trained to approximate $m_{0|t,y}$. However, when only an unconditional denoiser $m_{0|t} = \mathbb{E}[x_0 \mid x_t]$ is available, training-free conditional inference methods are employed.

**Diffusion Posterior Sampling (DPS) and its variants**. Given unconditional denoiser $\mathbb{E}[x_0 \mid x_t]$, training-free conditional inference methods enable the approximation of the ideal conditional denoiser $\mathbb{E}[x_0 \mid x_t, y]$ (Pokle et al., 2023):

$$\mathbb{E}[x_0 \mid x_t, y] = \mathbb{E}[x_0 \mid x_t] + \frac{\sigma_t^2}{\alpha_t} \nabla_{x_t} \log p(y \mid x_t). \tag{3}$$

Since $\nabla_{x_t} \log p(y \mid x_t)$ is generally intractable, various approaches have been developed to approximate it, such as heuristic approximation (Fei et al., 2023).

For linear inverse problems, where the forward model is given by: $y = Hx_0 + \epsilon, \epsilon \sim \mathcal{N}(0, \sigma_y^2 \mathbb{I})$. Tweedie Moment Projected Diffusion (TMPD) (Boys et al., 2023) provides a more accurate approximation. TMPD assumes $p(x_0 \mid x_t)$ as a Gaussian: $p(x_0 \mid x_t) \approx \mathcal{N}(m_{0|t}, C_{0|t})$, where $m_{0|t}(x_t) := \mathbb{E}[x_0 \mid x_t]$ is the ideal unconditional denoiser, $C_{0|t}(x_t) := \mathbb{E}[(x_0 - m_{0|t})(x_0 - m_{0|t})^T \mid x_t]$ is the covariance of $x_0 \mid x_t$. Then the posteroir mean $\mathbb{E}[x_0 \mid x_t, y]$ admits an explicit closed-form solution:

$$\mathbb{E}[x_0 \mid x_t, y] = m_{0|t} + C_{0|t}H^T(HC_{0|t}H^T + \sigma_y^2 \mathbb{I})^{-1}(y - Hm_{0|t}) \tag{4}$$

The covariance $C_{0|t}$ could be calculated via gradient go through the denoiser: $C_{0|t} = \frac{\sigma_t^2}{\alpha_t} \nabla_{m_{0|t}}(x_t)$. Since calculating the gradient with respect to $m_{0|t}$ is time-consuming, Yismaw et al. (2025) and Peng et al. (2024) try to find the optimal isotropic approximation of $C_{0|t}$, i.e., $C_{0|t} \approx r_t^2 \mathbb{I}$.

**Optimization based methods**. Unlike DPS guarantees that sampling is strictly from the conditional distribution, $p(x_0 \mid y)$, optimization-based approaches (Zhu et al., 2023; Li et al., 2024; Wang et al., 2024) place more emphasis on the alignment with the measurement and the prior, which takes the following iteration:

$$x_0^* = \arg\min_{x_0} \left\| x_0 - m_{0|t} \right\|^2 + \frac{1}{\eta_t} \left\| y - \mathcal{H}(x_0) \right\|^2, \tag{5a}$$

$$x_{t-\Delta t} \sim \mathcal{N}\left(a_t x_0^* + b_t x_t, \, c_t^2\right). \tag{5b}$$

where $\eta_t$ is a manually designed hyperparameter and $\mathcal{H}(\cdot)$ is the nonlinear forward operator. The iteration of optimization based methods could be seen as replacing $m_{0|t,y}$ in Eq. (2) to $x_0^*$ in Eq. (5a).

## 3 METHODOLOGY

For optimization based methods, the data-consistency loss with respect to the measurement $y$ is treated uniformly across all directions of the measurement space. However, for inverse problems it is often advantageous to introduce a weighting matrix that reflects the geometry of the forward operator (Tarantola, 2005). To this end, we propose Measurement-Aligned Sampling (MAS), which incorporates such a weighting into the optimization. As we demonstrate in Sec. 5, this alignment leads to significant improvements in reconstruction quality.

---

**Algorithm 1** Measurement-Aligned Sampling (MAS) for inverse problems.

---

1: **Input:** measurement $y$, forward operator $H(\cdot)$, pretrained DM $\epsilon_\theta(\cdot)$, number of diffusion step $N$, diffusion schedule $\alpha_t$ and $\sigma_t$, objective parameters $\eta_1, \eta_2$.
2: **Initialization**: $x_N \sim \mathcal{N}(0, \mathbb{I})$
3: **for** $n = N$ to 1 **do**
4:     $\hat{x}_0 \leftarrow [x_n - \sigma_n \epsilon_\theta(x_n, n)]/\alpha_n$                  ▷ Obtain predicted data $\mathbb{E}[x_0 \mid x_n]$
5:     $x_0' = Y^{-1}[\hat{x}_0 + H^\mathsf{T} W^{-1} y]$             ▷ Calculating posterior mean $\mathbb{E}[x_\epsilon \mid x_n, y]$
6:     $x_{n-1} \sim \mathcal{N}(\alpha_{n-1} x_0', \sigma_{n-1} \mathbb{I})$                           ▷ Forward diffusion step
7: **end for**
8: **Output** $x_0$

---

## 3.1 Measurement Aligned Sampling

In this work, we generalize the objective in Eq. (5a) as

$$x_0^* = \arg\min_{x_0} \|x_0 - m_{0|t}\|^2 + \|y - Hx_0\|_{W^{-1}}^2. \tag{6}$$

where $W := \eta_1 HH^\mathsf{T} + \eta_2 \mathbb{I}$ (with $\eta_1 \geq 0, \eta_2 \geq 0$) is the weighted matrix and serves as a metric that balances measurement fidelity and prior regularization, where $\|z\|_A^2 = z^T A z$.

$\eta_1$ and $\eta_2$ are the primary hyperparameters of MAS that need to design. When $\eta_1 = 0$ and $\eta_2 > 0$, corresponding to the classical Tikhonov (ridge) regularization, where $\eta_2$ controls the trade-off between fitting the measurements $y$ and staying close to the prior $m_{0|t}$. When $\eta_1 > 0$ and $\eta_2 = 0$, the data term becomes weighted by $(HH^\mathsf{T})^{-1}$, a Mahalanobis-type distance that emphasizes alignment along directions where $H$ is weak (small singular values), thereby regularizing ill-posed components of the inverse problem. *The balance between $\eta_1$ and $\eta_2$ is crucial for reconstruction performance*, as we show in our experiments.

Finally, Eq. (6) admits a unique closed-form solution obtained by setting the gradient to zero:

$$x_0^* = Y^{-1}[m_{0|t} + H^\mathsf{T} W^{-1} y]$$
$$\text{where} \quad W := \eta_1 HH^\mathsf{T} + \eta_2 \mathbb{I}, \qquad Y := \mathbb{I} + H^\mathsf{T} W^{-1} H, \tag{7}$$

In practice, computing the inverse $W^{-1}$ and $Y^{-1}$ in Eq. (7) naively can be computationally expensive. Instead, one can employ singular value decomposition (SVD) for more efficient computation; see Sec. B.2 for details.

*Remark* 1 (**Connection with DDNM** (Wang et al., 2022)). As $\eta_2 = 0$ and $\eta_1 \to 0$, $x_0^* \to \tilde{x}_0^{\text{DDNM}} := m_{0|t} + H^\dagger(y - Hm_{0|t})$. Thus, in this limiting case, MAS recovers DDNM.

*Remark* 2 (**Connection with optimization methods**). For the case where $\eta_1 = 0, \eta_2 > 0$, Eq. (6) reproduces optimization approaches, such as Resample (Song et al., 2023a), DiffPIR (Zhu et al., 2023), DCDP (Li et al., 2024), DMPlug (Wang et al., 2024).

## 3.2 Probabilistic interpretation

We can interpret $x_0^*$ in Eq. (7) as $\mathbb{E}[x_\epsilon \mid x_t, y]$, where $x_\epsilon \approx x_0$ with perturbation variance $\sigma_\epsilon^2$ chosen to be sufficiently small so that $p(x_0 \mid x_\epsilon) \approx \mathcal{N}(x_\epsilon, \sigma_\epsilon^2 \mathbb{I})$. Given the measurement model $p(y \mid x_0) = \mathcal{N}(Hx_0, \sigma_y^2 \mathbb{I})$ and conditional $p(x_0 \mid x_\epsilon) = \mathcal{N}(x_\epsilon, \sigma_\epsilon^2 \mathbb{I})$, the induced distribution over the measurement conditioned on $x_\epsilon$ takes the explicit form:

$$p(y \mid x_\epsilon) = \mathcal{N}\left(Hx_\epsilon,\ \sigma_y^2 \mathbb{I} + \sigma_\epsilon^2 HH^\mathsf{T}\right). \tag{8}$$

Notably, the likelihood $p(y \mid x_\epsilon)$ shares the same mean as $p(y \mid x_0)$, but with a generalized variance inflated by a term depending on $H$. Since both $p(y \mid x_\epsilon)$ and $p(y \mid x_0)$ are Gaussian, the posterior distribution admits a closed-form expression. In particular, the posterior mean $\mathbb{E}[x_0 \mid x_t, y]$ can be computed via Bayesian linear regression, as stated in Prop. 3.1.

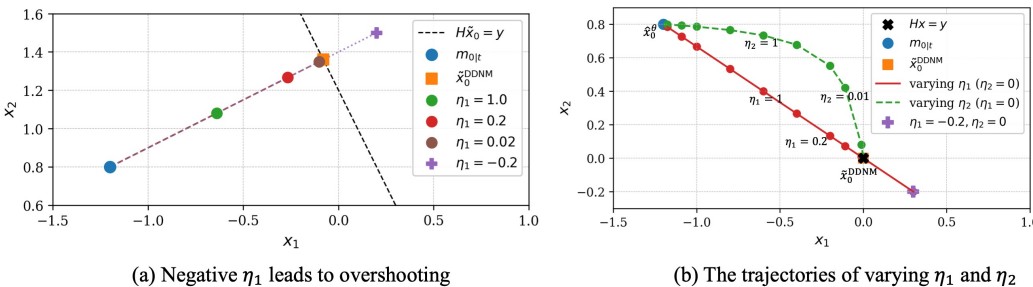

(a) Negative $\eta_1$ leads to overshooting

(b) The trajectories of varying $\eta_1$ and $\eta_2$

Figure 2: 2D illustration of the influence of parameters $\eta_1$ and $\eta_2$. Dots represent $x_0^*$, calculated via Eq. (7). (a) Parameter $\eta_1$ controls the trade-off between $m_{0|t}$ and $\tilde{x}_0^{\text{DDNM}}$: as $\eta_1 \to \infty$, the posterior mean $x_0^*$ approaches $m_{0|t}$; as $\eta_1 \to 0$, it converges to $\tilde{x}_0^{\text{DDNM}}$. (b) Adjusting $\eta_2$ alters the posterior trajectory differently from varying $\eta_1$.

**Proposition 3.1** (Bayesian Linear Regression). *Suppose $p(y \mid x_\epsilon) = \mathcal{N}(Hx_\epsilon, R)$, $R := \sigma_y^2\mathbb{I} + \sigma_\epsilon^2 HH^\mathsf{T}$ and $p(x_\epsilon \mid x_t) \approx \mathcal{N}(m_{0|t}, C_{0|t})$. Then the posterior is Gaussian, with mean given by*

$$\mathbb{E}[x_\epsilon \mid x_t, y] = \left(C_{0|t}^{-1} + H^\mathsf{T} R^{-1} H\right)^{-1}\left(C_{0|t}^{-1} m_{0|t} + H^\mathsf{T} R^{-1} y\right). \tag{9}$$

As we set $C_{0|t} = r_t^2\mathbb{I}$, $\eta_1 := \sigma_\epsilon^2/r_t^2$ and $\eta_2 := \sigma_y^2/r_t^2$, $\mathbb{E}[x_\epsilon \mid x_t, y]$ in Eq. (9) is equivalent to $x_0^*$ in Eq. (7), which provides a probabilistic perspective for MAS. Note $r_t$ and $\sigma_\epsilon$ serve only to motivate $\eta_1$ and $\eta_2$ and they are not additional hyperparameters to be tuned in practice.

*Remark* 3 (**Connection to TMPD (Boys et al., 2023)**). Setting $\sigma_\epsilon = 0$ reduces the posterior mean in Eq. (9) to that of TMPD in Eq. (4).

*Remark* 4 (**'Overshooting' trick**). Theoretically, $\eta_1 \geq 0$ since $\eta_1 := \sigma_\epsilon^2/r_t^2$, however, the posterior mean in Eq. (7) allows negative $\eta_1$. As illustrated in Fig. 2, negative $\eta_1$ produces an overshooting effect, drawing $x_0^*$ even further toward alignment with the measurement $y$ than prescribed by DDNM. Interestingly, in our experiments this overshooting effect leads to improved reconstruction quality. A more detailed discussion is provided in Sec. B.2.

## 4 MAXIMIZING THE CONSISTENCY FOR NOISY INVERSE PROBLEMS

### 4.1 WHY PREVIOUS METHODS FAILED TO MAXIMIZE THE CONSISTENCY?

DDNM highlighted that calculating the posterior sampling $\tilde{x}_0^{\text{DDNM}} = m_{0|t} + H^\dagger(y - Hm_{0|t})$ can inadvertently introduce additional noise into $x_t$, if $y$ is noisy. For instance, consider a simple forward model: $y = x_0 + \epsilon_y$, where both $H$ and $H^\dagger$ are identity matrix, i.e., $H = H^\dagger = \mathbb{I}$, then $\tilde{x}_0^{\text{DDNM}} = y = x_0 + \epsilon_y$. Here $\epsilon_y$ is the additional noise introduced to $\tilde{x}_0^{\text{DDNM}}$, and will be further introduced into $x_{t-\Delta t}$. We argue that this issue is not unique to DDNM, but may also arise in TMPD (Boys et al., 2023), DAPS (Zhang et al., 2024), as well as in optimization-based methods (Zhu et al., 2023; Li et al., 2024).

For MAS and under this same example, $y = x_0 + \epsilon_y$, calculating $x_0^*$ (Eq. (7)) yields

$$x_0^* = m_{0|t} + \frac{y - m_{0|t}}{\eta_1 + \eta_2 + 1} = m_{0|t} + \frac{x_0 - m_{0|t}}{\eta_1 + \eta_2 + 1} + \frac{\epsilon_y}{\eta_1 + \eta_2 + 1}, \tag{10}$$

where $\epsilon_y/(\eta_1 + \eta_2 + 1)$ is the additional noise introduced to $x_0^*$. A delicate balance arises from the fact that increasing either $\eta_1$ or $\eta_2$ will not only reduce the influence of the (unknown) noise term $\epsilon_y$, but also reduce the consistency with the measurement $y$ in general. To address this issue, we propose two approaches for addressing known Gaussian noise (Sec. 4.2) and unknown noise Sec. 4.3.

### 4.2 ADDRESSING GAUSSIAN NOISE WITH KNOWN VARIANCE

To handle Gaussian noise with known variance and $H = \mathbb{I}$, we modify Eq. (10) and Eq. (2) as:

Figure 3: The sample process of solving inverse problems with unknown noise, where $\hat{x}_0^\theta \approx m_{0|t}$ is the denoising output. Here we set $\eta_1 = 0$ and $\eta_2 = 0.5a_t/c_t$.

$$x_0^* = m_{0|t} + \lambda_t \frac{y - m_{0|t}}{\eta_1 + \eta_2 + 1}, \quad x_{t-\Delta t} \sim \mathcal{N}(a_t \tilde{x}_0 + b_t x_t, \gamma_t \mathbb{I}). \tag{11}$$

Here $\lambda_t$ and $\gamma_t$ are two parameters that can control the total noise introduced to $x_{t-\Delta t}$. In our work, we adopt similar two principles as DDNM+ (Wang et al., 2022) for handling Gaussian noise: (i) the total noise introduced in $x_{t-\Delta t}$ should be $\mathcal{N}(0, c_t^2\mathbb{I})$ to conform to the correct distribution of $x_{t-\Delta t}$ in Eq. (2); (ii) $\lambda_t$ should be as close to 1 as possible to maximize the preservation of $x_0^*$. As $\epsilon_y \sim \mathcal{N}(0, \sigma_y^2\mathbb{I})$, principle (i) and principle (ii) are equivalent to:

$$\left(\frac{a_t \lambda_t \sigma_y}{\eta_1 + \eta_2 + 1}\right)^2 + \gamma_t = c_t^2, \quad \lambda_t = \begin{cases} 1, & c_t \geq \dfrac{a_t \sigma_y}{\eta_1 + \eta_2 + 1} \\ \dfrac{c_t(\eta_1 + \eta_2 + 1)}{a_t \sigma_y}, & c_t < \dfrac{a_t \sigma_y}{\eta_1 + \eta_2 + 1} \end{cases}. \tag{12}$$

Derivations for more general forms of $H$ can be found in Sec. B. Note that the revision does not introduce additional parameters.

### 4.3 Addressing unknown noise and non-differentiable measurements

**Addressing unknown noise or non-Gaussian noise.** When the measurement noise $\sigma_y$ is non-Gaussian or unknown, it becomes difficult to ensure that the total noise in $x_{t-\Delta t}$ follows the desired distribution $\mathcal{N}(0, c_t^2\mathbb{I})$, To address this, we continue to sample $x_{t-\Delta t}$ using Eq. (2). Next, the noise introduced to $x_{t-\Delta t}$ is the sum of two components:

$$\epsilon_{\text{ng}} = (a_t \lambda_t \epsilon_y)/(\eta_1 + \eta_2 + 1), \quad \epsilon_g \sim \mathcal{N}(0, c_t^2\mathbb{I}). \tag{13}$$

Here $\epsilon_{\text{ng}}$ is related to the noise introduced by unknown noise $\epsilon_y$, while $\epsilon_g$ is the noise added by the diffusion process. To minimize the effect of unknown noise $\epsilon_{\text{ng}}$, it is desirable for $\eta_1 + \eta_2 + 1$ to be sufficiently large. However, smaller values of $\eta_1$ and $\eta_2$ result in better consistency with the measurement $y$, as illustrated in Fig. 2. To balance this trade-off, we propose using a small $\eta_1 + \eta_2$ during the early stages of sampling to fully exploit measurement information. As sampling progresses, $\eta_1 + \eta_2$ should be gradually increased to suppress the impact of $\epsilon_{\text{ng}}$. The underlying intuition is that, in the early sampling stage, $x_t$ is still highly noisy and $a_t \approx 0$, so the influence of $\epsilon_{\text{ng}}$ is negligible even when $\eta_1 + \eta_2$ is small. As shown in Fig. 3, $x_0^*$ is initially more aligned with the degraded observation, but progressively shifts toward $m_{0|t}$ as sampling evolves.

For a general degradation operator $H$, we recommend setting $\eta_2 = ka_t/c_t$, where $k$ is a constant determined by the characteristics of the introduced noise. The rationale behind this design choice is further detailed in Sec. B.

**Addressing non-differentiable measurements.** For solving inverse problems with non-differentiable measurements such as JPEG restoration and quantization, the degraded images can be viewed as "noisy images" with unknown noise, modeled by $y = x + \epsilon_y$. In these scenarios, our proposed strategy naturally extends by treating the unknown degradations as implicit noise.

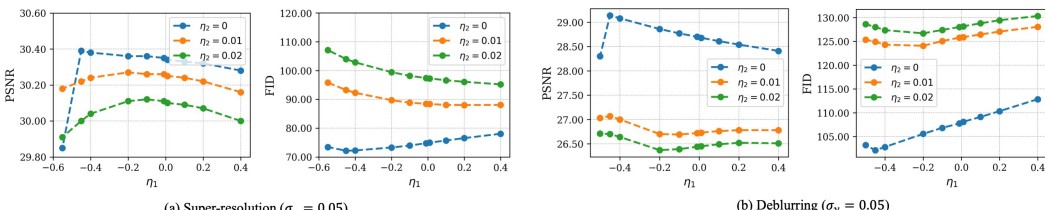

Figure 4: Ablation study of $\eta_1$ and $\eta_2$ on solving super-resolution and deblurring. We set NFE=20 for all tasks.

## 5 EXPERIMENTS

### 5.1 EXPERIMENTAL SETUP

**Dataset**. We evaluate the effectiveness of our proposed approach on FFHQ $256 \times 256$ (Karras et al., 2019) and ImageNet $256 \times 256$ (Deng et al., 2009). Following DAPS (Zhang et al., 2024), we test on the same subset of 100 images for both datasets.

**Pretrained models and baselines**. We utilize the pre-trained checkpoint (Chung et al., 2022a) on the FFHQ dataset and the pre-trained checkpoint (Dhariwal & Nichol, 2021) on the Imagenet dataset. We compare our methods with the following baselines: DCDP (Li et al., 2024), FPS (Dou & Song, 2024), DiffPIR (Zhu et al., 2023), DDNM (Wang et al., 2022), DDRM (Kawar et al., 2022a), ΠGDM (Song et al., 2023b), RedDiff (Mardani et al., 2023), DAPS (Zhang et al., 2024).

**Metrics**. Following previous work (Chung et al., 2022a; Kawar et al., 2022a), we report Fréchet Inception Distance (FID) (Heusel et al., 2017), Learned Perceptual Image Patch Similarity (LPIPS) (Zhang et al., 2018), Peak Signal-to-Noise Ratio (PSNR), and Structural SIMilarity index (SSIM).

**Tasks**. (1) We evaluate performance on the following linear inverse problems: super-resolution (bicubic filter), deblurring (uniform kernel of size 9), inpainting (with a box mask), inpainting (with a 70% random mask), and colorization. (2) We consider two unknown noise types: salt-and-pepper noise (10% pixels set randomly to $\pm 1$) and periodic noise (sinusoidal pattern with amplitude 0.2 and frequency 5). (3) We address JPEG restoration with quality factors QF = 2 and QF = 5. (4). For quantization, we consider the challenging case of 2-bit quantization.

### 5.2 ABLATION STUDY

**Ablation Study on $\eta_1$ and $\eta_2$.** We conduct ablation studies on parameters $\eta_1$ and $\eta_2$ using two inverse problems: super-resolution (noise-free, $\epsilon_y = 0$) and deblurring (noisy, $\epsilon_y \sim \mathcal{N}(0, \sigma_y^2 \mathbb{I})$). Results presented in Fig. 4 demonstrate that for noise-free super-resolution, the highest PSNR and lowest FID scores are achieved by setting $\eta_2 = 0$ and a negative $\eta_1 = -0.45$. This indicates that appropriate "overshooting" enhances restoration quality. For the noisy deblurring task, negative $\eta_2$ yields an improvement of more than 0.5 in PSNR and a reduction of over 5 in FID, further confirming the benefit of overshooting.

### 5.3 IMAGE RESTORATION

**Inverse problems with Gaussian noise (known variance).** Quantitative results for inverse problems with Gaussian noise of known variance are shown in Table 2. MAS consistently demonstrates superior performance across most tasks, notably achieving significantly higher PSNRs. The table summarizes 5 tasks, 4 restoration quality metrics, and 2 datasets, resulting in a total of 40 evaluations. MAS demonstrates superior performance in 29 out of the 40 cases. Notably, MAS achieves improvements of more than 1 dB in 5 out of 10 instances.

**Inverse Problems with Non-Gaussian Noise (Unknown Strength).** Quantitative evaluations for linear inverse problems with unknown, non-Gaussian noise are presented in Table 3. MAS consistently outperforms baseline methods, highlighting the effectiveness of our approach in handling unknown noise conditions.

Table 2: Quantitative evaluation of solving image restoration FFHQ (left) and ImageNet (right), with Gaussian noise (known variance, $\sigma_y = 0.05$).

| Task | Method | FFHQ | | | | ImageNet | | | |
|------|--------|------|------|------|------|----------|------|------|------|
| | | PSNR ↑ | SSIM ↑ | LPIPS ↓ | FID ↓ | PSNR ↑ | SSIM ↑ | LPIPS ↓ | FID ↓ |
| SR 4× | DPS | 25.86 | 0.753 | 0.269 | 81.07 | 21.13 | 0.489 | 0.361 | 106.32 |
| | DDRM | 26.58 | 0.782 | 0.282 | 79.25 | 22.62 | 0.521 | 0.324 | 103.85 |
| | DDNM | 28.03 | 0.795 | 0.197 | 64.62 | 23.96 | 0.604 | 0.475 | 98.62 |
| | DCDP | 28.66 | 0.807 | 0.178 | 53.81 | – | – | – | – |
| | FPS-SMC | 28.42 | 0.813 | 0.204 | 49.25 | 24.82 | 0.703 | 0.313 | 97.51 |
| | DiffPIR | 26.64 | – | 0.260 | 65.77 | 23.18 | – | 0.371 | 106.32 |
| | RED-Diff | 28.63 | 0.748 | 0.288 | 126.78 | 25.43 | 0.639 | 0.336 | 153.37 |
| | DAPS | 29.07 | 0.818 | 0.177 | 51.44 | 25.89 | 0.694 | 0.276 | 83.57 |
| | MAS | **30.56** | **0.865** | **0.131** | 61.38 | **27.20** | **0.751** | **0.215** | 88.61 |
| Inpaint (Box) | DPS | 22.51 | 0.792 | 0.209 | 61.27 | 18.94 | 0.722 | 0.257 | 126.52 |
| | DDRM | 22.26 | 0.801 | 0.207 | 78.62 | 18.63 | 0.733 | 0.254 | 116.37 |
| | DDNM | 24.47 | 0.837 | 0.235 | 46.59 | 21.64 | 0.748 | 0.319 | 103.97 |
| | DCDP | 23.89 | 0.760 | 0.163 | 45.23 | – | – | – | – |
| | FPS-SMC | 24.86 | 0.823 | 0.146 | 48.34 | 22.16 | 0.726 | 0.208 | 111.58 |
| | RED-Diff | 24.68 | 0.767 | 0.175 | 86.78 | 21.32 | 0.728 | 0.247 | 123.55 |
| | DAPS | 24.07 | 0.814 | 0.133 | 43.10 | 21.43 | 0.725 | 0.214 | 109.85 |
| | MAS | **24.95** | **0.879** | **0.082** | **37.67** | 21.15 | **0.817** | **0.168** | **95.96** |
| Inpaint (Random) | DPS | 25.46 | 0.823 | 0.203 | 69.20 | 23.52 | 0.745 | 0.297 | 87.53 |
| | DDNM | 29.91 | 0.817 | 0.121 | 44.37 | 31.16 | 0.841 | 0.191 | 63.84 |
| | DCDP | 30.69 | 0.842 | 0.142 | 52.51 | – | – | – | – |
| | FPS-SMC | 28.21 | 0.823 | 0.261 | 61.23 | 24.52 | 0.701 | 0.316 | 79.12 |
| | RED-Diff | 29.73 | 0.814 | 0.200 | 104.19 | 27.04 | 0.753 | 0.226 | 92.24 |
| | DAPS | 31.12 | 0.844 | 0.098 | 32.17 | 28.44 | 0.775 | 0.135 | 54.25 |
| | MAS | **33.10** | **0.923** | **0.073** | 34.75 | **29.05** | **0.838** | **0.113** | **30.19** |
| Deblurring (Uniform) | DDNM | 26.58 | 0.704 | 0.210 | 68.83 | 25.69 | 0.630 | 0.261 | 83.63 |
| | DDRM | 29.19 | 0.835 | 0.172 | 87.12 | 26.31 | 0.711 | 0.267 | 118.36 |
| | DAPS | 28.92 | 0.758 | 0.204 | 76.57 | 25.43 | 0.616 | 0.293 | 103.55 |
| | MAS | **30.58** | **0.857** | 0.174 | 103.88 | 26.25 | 0.700 | 0.295 | 141.58 |
| Color | DDNM | 24.83 | 0.868 | 0.244 | 85.15 | 22.57 | 0.884 | 0.271 | 87.48 |
| | DDRM | 23.27 | 0.881 | 0.250 | 100.48 | 21.12 | 0.819 | 0.346 | 103.39 |
| | RED-Diff | 24.21 | 0.785 | 0.304 | 107.64 | 22.18 | 0.782 | 0.368 | 104.40 |
| | DAPS | 23.92 | 0.825 | 0.263 | 88.09 | 22.13 | 0.830 | 0.323 | 89.30 |
| | MAS | 24.23 | **0.919** | **0.187** | **72.33** | **22.66** | **0.886** | **0.258** | **83.17** |

Table 3: Quantitative evaluation of solving linear inverse problems with non-Gaussian noise (unknown strength).

| Task | Method | Salt peper noise | | | | Periodic noise | | | | Poisson noise | | | |
|------|--------|------|------|------|------|------|------|------|------|------|------|------|------|
| | | PSNR ↑ | SSIM ↑ | LPIPS ↓ | FID ↓ | PSNR ↑ | SSIM ↑ | LPIPS ↓ | FID ↓ | PSNR ↑ | SSIM ↑ | LPIPS ↓ | FID ↓ |
| SR 8× | DDNM | 13.02 | 0.289 | 0.710 | 377.54 | 18.61 | 0.492 | 0.495 | 268.36 | 18.79 | 0.392 | 0.653 | 349.88 |
| | DDRM | 16.06 | 0.506 | 0.629 | 351.69 | 19.74 | 0.545 | 0.463 | 218.38 | 23.47 | 0.651 | 0.417 | 147.70 |
| | ΠGDM | 17.36 | 0.476 | 0.569 | 309.73 | 18.12 | 0.449 | 0.434 | 163.41 | 17.33 | 0.315 | 0.640 | 300.97 |
| | RED-Diff | 14.21 | 0.357 | 0.668 | 342.09 | 19.47 | 0.596 | 0.416 | 224.07 | 19.62 | 0.450 | 0.636 | 306.88 |
| | MAS (ours) | **20.05** | **0.605** | **0.390** | **129.80** | **20.10** | **0.591** | 0.395 | **137.57** | **23.69** | **0.700** | **0.304** | **115.16** |
| Inpaint (Box) | DDNM | 15.55 | 0.248 | 0.533 | 247.99 | 18.60 | 0.621 | 0.341 | 147.80 | 21.30 | 0.483 | 0.350 | 155.41 |
| | DDRM | 20.27 | 0.599 | 0.350 | 142.01 | 18.74 | 0.589 | 0.423 | 199.14 | 21.10 | 0.734 | 0.263 | 115.24 |
| | ΠGDM | 19.30 | 0.665 | 0.297 | 100.07 | 18.32 | 0.601 | 0.349 | 150.49 | 22.10 | 0.551 | 0.318 | 131.46 |
| | RED-Diff | 15.75 | 0.287 | 0.523 | 255.85 | 19.13 | 0.638 | 0.338 | 159.99 | 21.93 | 0.522 | 0.339 | 175.14 |
| | MAS (ours) | **22.78** | **0.723** | **0.244** | **90.15** | **19.13** | 0.581 | 0.407 | **138.56** | **23.26** | **0.746** | **0.253** | **102.11** |

**Inverse problems with non-differentiable measurements.** MAS is also capable of solving inverse problems with non-differentiable measurements, such as JPEG restoration and quantization. Results in Table 4 and Fig. 5 show that MAS achieves state-of-the-art performance without relying on the forward operator or knowledge of the degradation strength.

**Computational time analysis**. The computational efficiency of MAS is comparable to DDNM and substantially higher than DAPS. For example, on the SR task using the FFHQ-256 dataset with 200 NFEs, the non-parallel single-image sampling time for both DDNM and MAS is only 8 seconds, whereas DAPS requires 67 seconds (test were conducted on the same NVIDIA A6000 GPU).

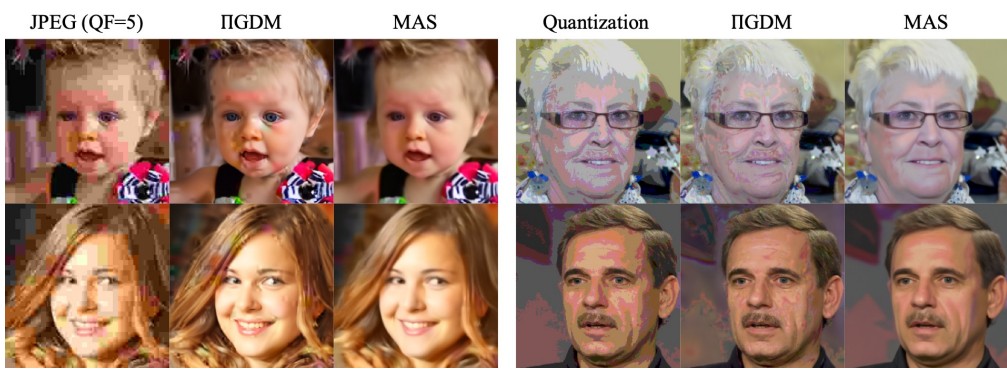

Figure 5: Results on JPEG (QF=5) and quantization restoration.

Table 4: Quantitative evaluation of solving JPEG restoration and Quantization. We set $k = 1.0$ for QF = 5 and $k = 3.0$ for QF = 2, and $k = 0.5$ for quantization. Both $\Pi$GDM and MAS use NFE = 20, which yields the best performance (among NFE = 20 and NFE = 100). Notably, our method (MAS) does not require access to the forward operator or the strength of degration.

| Method | JPEG Restoration (QF = 5) | | | | JPEG Restoration (QF = 2) | | | | Quantization (number of bits = 2) | | | |
|---|---|---|---|---|---|---|---|---|---|---|---|---|
| | PSNR ↑ | SSIM ↑ | LPIPS ↓ | FID ↓ | PSNR ↑ | SSIM ↑ | LPIPS ↓ | FID ↓ | PSNR ↑ | SSIM ↑ | LPIPS ↓ | FID ↓ |
| $\Pi$GDM | 25.78 | 0.750 | 0.241 | 89.82 | 22.92 | 0.653 | 0.314 | 112.27 | 29.98 | 0.823 | 0.185 | 124.57 |
| MAS (ours) | **26.30** | **0.787** | **0.281** | 101.24 | **23.72** | **0.772** | **0.335** | 114.85 | 28.97 | **0.837** | **0.196** | **69.61** |

## 6 RELATED WORK

Diffusion models have also been successfully applied to linear inverse problems, including, compressed-sensing MRI (CS-MRI), and computed tomography (CT) (Kadkhodaie & Simoncelli, 2021; Song et al., 2020b; Chung et al., 2022b; Kawar et al., 2022a; Song et al., 2021). They have also been extended to non-linear inverse problems such as Fourier phase retrieval, nonlinear deblurring, HDR, and JPEG restoration (Chung et al., 2022a; Song et al., 2023b; Chung et al., 2023; Mardani et al., 2023).

Methods to solve inverse problems include linear projection methods (Wang et al., 2022; Kawar et al., 2022a; Dou & Song, 2024), Monte Carlo sampling (Wu et al., 2023; Phillips et al., 2024), variational inference (Feng et al., 2023; Mardani et al., 2023; Janati et al., 2024), optimization-based approaches (Song et al., 2023a; Zhu et al., 2023; Li et al., 2024; Wang et al., 2024; Alkhouri et al., 2024; He et al., 2023), and Diffusion Posterior Sampling (DPS) (Zhang et al., 2024; Chung et al., 2022a; Song et al., 2023c; Yu et al., 2023; Rout et al., 2024; Yang et al., 2024; Bansal et al., 2023; Boys et al., 2023; Song et al., 2023b; Ho & Salimans, 2022). Besides, InverseBench (Zheng et al., 2025) presents a benchmark for critical scientific applications, which present structural challenges that differ significantly from natural image restoration tasks.

## 7 CONCLUSION

MAS improves upon existing methods by explicitly aligning the sampling process with measurement data, offering a broader optimization perspective that generalizes approaches like DDNM and DAPS. Beyond the noise-free case, MAS can be extended to: (1) known Gaussian noise, (2) unknown or non-Gaussian noise through adaptive parameterization, and (3) non-differentiable degradations (e.g., JPEG) by decoupling the forward operator from sampling. Extensive experiments show that MAS consistently outperforms state-of-the-art methods across a wide range of inverse problems. While MAS can handle non-differentiable measurements like JPEG restoration, it does not support general non-linear inverse problems, it's also promising to 'calibrate' the noise introduced into $x_t$, such that maximizing the consistency to measurement.

## REPRODUCIBILITY STATEMENT

All code and instructions necessary to reproduce our experiments are anonymously available at `https://anonymous.4open.science/r/MAS_linear-8C3C`. We provide a PyTorch-like implementation of the calculation of $x_0^*$ in Eq. (7), included in Sec. E.

## ETHICS STATEMENT

This work does not present any foreseeable ethical issues.

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

## A PROOFS

### A.1 PROOF OF PROP. 3.1.

*Proof.* Let $x \equiv x_\epsilon$. The prior and likelihood are

$$p(x \mid x_t) = \mathcal{N}(m_{0|t},\, C_{0|t}), \qquad p(y \mid x) = \mathcal{N}(Hx,\, R),$$

with $R = \sigma_y^2 I_m + \sigma_\epsilon^2 H H^\top$. Denote $m := m_{0|t}$ and $C := C_{0|t}$.

The posterior is, up to normalization,

$$p(x \mid x_t, y) \propto \exp\left(-\tfrac{1}{2}(x-m)^\top C^{-1}(x-m) - \tfrac{1}{2}(y-Hx)^\top R^{-1}(y-Hx)\right).$$

Expanding the exponent and collecting terms in $x$ gives

$$-\tfrac{1}{2}\left[x^\top C^{-1}x - 2x^\top C^{-1}m + m^\top C^{-1}m + x^\top H^\top R^{-1}Hx - 2x^\top H^\top R^{-1}y + y^\top R^{-1}y\right]$$

$$= -\tfrac{1}{2}\left[x^\top(C^{-1} + H^\top R^{-1}H)x - 2x^\top(C^{-1}m + H^\top R^{-1}y)\right] + \text{(terms independent of } x).$$

This is the quadratic form of a Gaussian in $x$ with precision

$$\Lambda \;=\; C^{-1} + H^\top R^{-1}H,$$

and natural parameter

$$\eta \;=\; C^{-1}m + H^\top R^{-1}y.$$

Therefore the posterior is Gaussian $\mathcal{N}(\mu_{\text{post}},\, \Sigma_{\text{post}})$ with

$$\Sigma_{\text{post}} = \Lambda^{-1} = \left(C^{-1} + H^\top R^{-1}H\right)^{-1}, \qquad \mu_{\text{post}} = \Sigma_{\text{post}}\,\eta = \left(C^{-1} + H^\top R^{-1}H\right)^{-1}\left(C^{-1}m + H^\top R^{-1}y\right).$$

Restoring the original notation gives equation 9. $\qquad\square$

### A.2 PROOF OF EFFIEIENT LINEAR SOLVES IN EQ. (38)

**Lemma A.1.** *Let $H \in \mathbb{R}^{m \times d}$ have (thin) singular–value decomposition $H = U\Sigma V^\mathsf{T}$ with orthogonal $U \in \mathbb{R}^{m \times m}$, $V \in \mathbb{R}^{d \times d}$ and $\Sigma = \mathrm{diag}(s_1, \ldots, s_r) \in \mathbb{R}^{m \times d}$, where $r = \mathrm{rank}(H)$ and $s_1 \geq \cdots \geq s_r > 0$. For any scalars $\eta_1 \geq 0$ and $\eta_2 > 0$ define*

$$W := \eta_1 H H^\mathsf{T} + \eta_2 \mathbb{I}, \qquad Y := \mathbb{I} + H^\mathsf{T} W^{-1} H.$$

*Then*

$$W^{-1} = U\, \mathrm{diag}\left(\tfrac{1}{\eta_1 s_i^2 + \eta_2}\right)_{i=1}^m U^\mathsf{T}, \qquad Y^{-1} = V\, \mathrm{diag}\left(\tfrac{1}{1 + s_i^2/(\eta_1 s_i^2 + \eta_2)}\right)_{i=1}^d V^\mathsf{T}. \tag{14}$$

*(When $i > r$ we set $s_i = 0$.)*

*Proof.* **(i) Inverting $W$.** Using the SVD,

$$W = \eta_1 U\Sigma\Sigma^\mathsf{T}U^\mathsf{T} + \eta_2 U\mathbb{I}U^\mathsf{T} = U\left(\eta_1 \Sigma\Sigma^\mathsf{T} + \eta_2\mathbb{I}\right)U^\mathsf{T}.$$

Because $U$ is orthogonal, $W^{-1}$ is obtained by inverting the diagonal middle matrix: $(\eta_1\Sigma\Sigma^\mathsf{T} + \eta_2\mathbb{I})^{-1} = \mathrm{diag}\left(\tfrac{1}{\eta_1 s_i^2 + \eta_2}\right)_{i=1}^m$. Substituting yields the first identity in equation 14.

**(ii) Inverting $Y$.** Write

$$Y = \mathbb{I} + H^\mathsf{T}W^{-1}H = V\Sigma^\mathsf{T}U^\mathsf{T}\left[U\,\mathrm{diag}\left(\tfrac{1}{\eta_1 s_i^2 + \eta_2}\right)U^\mathsf{T}\right]U\Sigma V^\mathsf{T} + \mathbb{I},$$

and simplify with $U^\mathsf{T}U = \mathbb{I}$:

$$Y = V\left[\Sigma^\mathsf{T}\mathrm{diag}\left(\tfrac{1}{\eta_1 s_i^2 + \eta_2}\right)\Sigma + \mathbb{I}\right]V^\mathsf{T}.$$

Because $\Sigma^\mathsf{T}\mathrm{diag}\left(\tfrac{1}{\eta_1 s_i^2 + \eta_2}\right)\Sigma$ is diagonal with $i^{\text{th}}$ entry $\tfrac{s_i^2}{\eta_1 s_i^2 + \eta_2}$, the bracketed matrix is diagonal and hence trivial to invert, giving the second identity in equation 14. $\qquad\square$

A.3 PROOF OF EQ. (6)

**Proposition A.2.** *Let $H \in \mathbb{R}^{m \times d}$, $\eta_1 \geq 0$ and $\eta_2 > 0$. Define*

$$W = \eta_1 H H^\mathsf{T} + \eta_2 \mathbb{I}, \qquad Y = \mathbb{I} + H^\mathsf{T} W^{-1} H.$$

*For any $y \in \mathbb{R}^m$ and $m_{0|t} \in \mathbb{R}^d$ consider the strictly convex quadratic*

$$\mathcal{L}(x_0) = \left\| x_0 - m_{0|t} \right\|_2^2 + \left\| y - H\tilde{x}_0 \right\|_{W^{-1}}^2, \qquad \|v\|_{W^{-1}}^2 = v^\mathsf{T} W^{-1} v.$$

*Its unique minimiser is*

$$\tilde{x}_0^* = Y^{-1}\big[ m_{0|t} + H^\mathsf{T} W^{-1} y \big]. \tag{15}$$

*Proof.* Expand $\mathcal{L}$ and take its gradient:

$$\nabla_{\tilde{x}_0} \mathcal{L} = 2\big( \tilde{x}_0 - m_{0|t} \big) - 2 H^\mathsf{T} W^{-1} \big( y - H\tilde{x}_0 \big).$$

Setting $\nabla_{\tilde{x}_0} \mathcal{L} = 0$ gives the *normal equation*

$$\big( \mathbb{I} + H^\mathsf{T} W^{-1} H \big) \tilde{x}_0 = m_{0|t} + H^\mathsf{T} W^{-1} y, \qquad \text{that is, } Y \tilde{x}_0 = m_{0|t} + H^\mathsf{T} W^{-1} y.$$

Because $\eta_2 > 0$ implies $W \succ 0$, we have $W^{-1} \succ 0$ and hence $Y = \mathbb{I} + H^\mathsf{T} W^{-1} H \succ 0$; thus $Y$ is invertible and equation 15 follows.

Finally, the Hessian of $\mathcal{L}$ is $2Y \succ 0$, so $\mathcal{L}$ is strictly convex and the stationary point equation 15 is indeed its unique global minimiser. $\qquad\square$

A.4 PROOF OF REMARK 1.

*Proof.* As $\eta_2 = 0$,

$$x_0^* = \big( \eta \mathbb{I} + H^\dagger H \big)^{-1} \big( \eta_1 m_{0|t} + H^\dagger y \big). \tag{16}$$

To analyze the limit as $\eta_1 \to 0$, decompose the space into two orthogonal components:

- The range (or row space) of $H$, on which $H^\dagger H$ acts as the identity.

- Its nullspace, on which $H^\dagger H$ is zero.

Let

$$P = H^\dagger H, \tag{17}$$

which is the orthogonal projection onto the row space of $H$. Then any vector $v$ can be decomposed as

$$v = Pv + (I - P)v. \tag{18}$$

Notice that $H^\dagger y$ lies in the row space (i.e. $P H^\dagger y = H^\dagger y$) and that $m_{0|t}$ can be decomposed as

$$m_{0|t} = P m_{0|t} + (I - P) m_{0|t}. \tag{19}$$

Since the eigenvalues of $P$ are 0 and 1, the matrix $\eta_1 I + P$ has eigenvalues $\eta_1$ (on the nullspace of $P$) and $1 + \eta_1$ (on the row space). Hence, its inverse acts as:

- Multiplication by $1/\eta_1$ on the nullspace,

- Multiplication by $1/(1 + \eta_1)$ on the row space.

Thus, we have

$$\big( \eta_1 I + P \big)^{-1} \big( H^\dagger y + \eta_1 m_{0|t} \big) = \frac{1}{1 + \epsilon} \big( H^\dagger y + \eta_1 P m_{0|t} \big) + \frac{1}{\eta_1} \big( \eta_1 (I - P) m_{0|t} \big). \tag{20}$$

Simplify this to obtain

$$\frac{1}{1 + \eta_1} H^\dagger y + \frac{\eta_1}{1 + \eta_1} P m_{0|t} + (I - P) m_{0|t}. \tag{21}$$

Now, taking the limit as $\eta_1 \to 0$:

- $\frac{1}{1+\eta_1} \to 1$,

- $\frac{\eta_1}{1+\eta_1} \to 0$.

Therefore, the limit becomes

$$\lim_{\eta_1 \to 0} x_0^* = H^\dagger y + (I - P)m_{0|t}. \tag{22}$$

Recalling that $P = H^\dagger H$, we rewrite this as

$$H^\dagger y + m_{0|t} - H^\dagger H\, m_{0|t} = m_{0|t} + H^\dagger\big(y - Hm_{0|t}\big). \tag{23}$$

Thus, in the limit where $\eta_1 \to 0$, we indeed have

$$x_0^* = m_{0|t} + H^\dagger\big(y - Hm_{0|t}\big). \tag{24}$$

This shows that, as the relative measurement noise $\epsilon$ becomes much smaller compared to the prior uncertainty $r_t$, the posterior expectation is the projection of $\hat{x}_0^\theta$ onto the subspace $\{x : Hx = y\}$. $\quad\square$

# B  ADDITIONAL METHOD DETAILS

## B.1  ADDRESSING GAUSSIAN NOISE

Consider noisy image restoration problems in the form of $y = Hx + \epsilon_y$, where $\epsilon_y$ is the added noise. Then the measurement $y$ can be decomposed to the sum of clean measurement $y^{\text{clean}} := Hx$ and measurement noise $\epsilon_y$. Calculating $x_0^*$ leads to:

$$x_0^* = Y^{-1}[m_{0|t} + H^\mathsf{T} W^{-1} y] \tag{25}$$
$$= m_{0|t} + (Y^{-1} - \mathbb{I})m_{0|t} + Y^{-1} H^T W^{-1} y \tag{26}$$

where $Y^{-1} H^T W^{-1} \epsilon_y$ is the extra noise introduced into $x_0^*$ and will be further introduced into $x_{t-\Delta t}$.

To address Gaussian noise with known variance, we modify Eq. (7) and Eq. (2) as:

$$x_0^* = m_{0|t} + \Sigma_t[(Y^{-1} - \mathbb{I})m_{0|t} + H^\mathsf{T} W^{-1} y] \tag{27}$$

$$x_{t-\Delta t} \sim \mathcal{N}(a_t \tilde{x}_0^{\text{pe}}(t, x, y) + b_t x_t, \Phi_t \mathbb{I}) \tag{28}$$

Then $x_0^*$ is:

$$x_0^* = m_{0|t} + \Sigma_t[(Y^{-1} - \mathbb{I})m_{0|t} + H^\mathsf{T} W^{-1} y] \tag{29}$$
$$\tag{30}$$
$$= \underbrace{m_{0|t} + \Sigma_t(Y^{-1} - \mathbb{I})m_{0|t} + Y^{-1} H^T W^{-1} y^{\text{clean}}}_{:= \tilde{x}_0^{\text{clean}}} + \Sigma_t Y^{-1} H^\mathsf{T} W^{-1} \epsilon_y \tag{31}$$

Then the iteration of the sampling process is:

$$x_{t-\Delta t} = a_t x_0^*(t, x, y) + b_t x_t + \epsilon_{\text{new}}, \quad \epsilon_{\text{new}} \sim \mathcal{N}(0, \Phi_t) \tag{32}$$
$$= a_t \tilde{x}_0^{\text{clean}} + b_t x_t + \underbrace{a_t \sigma_y Y^{-1} H^\mathsf{T} W^{-1} \epsilon_y}_{:= \epsilon_{\text{intro}}} + \epsilon_{\text{new}} \tag{33}$$

Suppose $\Sigma_t = V \mathrm{diag}\{\lambda_{t1}, \cdots, \lambda_{td}\} V^T$ $\Phi_t = V \mathrm{diag}\{\gamma_{t1}, \cdots \gamma_{td}\} V^T$. Then the introduced noise $\epsilon_{\mathrm{intro}} = a_t \sigma_y Y^{-1} H^\mathsf{T} W^{-1} \epsilon_y$ is still a Gaussian distribution: $\epsilon_{\mathrm{intro}} \sim \mathcal{N}(0, V D_t V^T)$, with $D_t = diag\{d_{t1}, \cdots, d_{td}\}$:

$$
d_{ti} = \begin{cases} \dfrac{a_t^2 \sigma_y^2 s_i^2 \lambda_{ti}^2}{\left[(\eta_1 + 1) s_i^2 + \eta_2\right]^2}, & s_i \neq 0, \\ 0, & s_i = 0, \end{cases} \tag{34}
$$

The choice of and $\Phi_t$ need to ensure the total noise injected to $x_{t-\Delta t}$ conforms the iteration in Eq. (2).

$$
\epsilon_{\mathrm{new}} + \epsilon_{\mathrm{intro}} \sim \mathcal{N}(0, c_t^2 \mathbb{I}) \tag{35}
$$

To construct $\epsilon_{\mathrm{new}}$, we define a new diagonal matrix $\Gamma_t (= diag\{\gamma_{t1}, \cdots \gamma_{td}\})$:

$$
\gamma_{ti} = \begin{cases} c_t^2 - \dfrac{a_t^2 \sigma_y^2 s_i^2 \lambda_{ti}^2}{\left[(\eta_1 + 1) s_i^2 + \eta_2\right]^2}, & s_i \neq 0, \\ c_t^2, & s_i = 0, \end{cases} \tag{36}
$$

Now we can yield $\epsilon_{\mathrm{new}}$ by sampling from $\mathcal{N}(0, V \Gamma_t V^T)$ to ensure that $\epsilon_{\mathrm{intro}} + \epsilon_{\mathrm{new}} \sim \mathcal{N}(0, c_t^2 \mathbb{I})$. We need to make sure $\lambda_{ti}$ guarantees the noise level of the introduced noise does not exceed the pre-defined noise level $c_t$, we also hope $\lambda_{t_i}$ as close as 1 as possible. Therefore,

$$
\lambda_{ti} = \begin{cases} 1, & c_t \geq \frac{a_t \sigma_y s_i}{(\eta_1 + 1) s_i^2 + \eta_2}, \\ \frac{c_t((\eta+1)s_i^2 + \eta_2)}{a_t \sigma_y s_i}, & c_t < \frac{a_t \sigma_y s_i}{(\eta_1 + 1) s_i^2 + \eta_2}, \\ 1, & s_i = 0. \end{cases} \tag{37}
$$

In practice, we found that setting $\sigma_y$ slightly larger than the true $\sigma_y$ is beneficial, possibly because the denoiser is more sensitive to excessive noise.

## B.2 EFFICIENT CALCULATION VIA SVD DECOMPOSITION

Let $H = U \Sigma V^\mathsf{T}$ with singular values $s_1, \ldots, s_n$. Then

$$
W^{-1} = U \mathrm{diag}\left(\frac{1}{\eta_1 s_i^2 + \eta_2}\right) U^\mathsf{T}, \qquad Y^{-1} = V \mathrm{diag}\left(\frac{1}{1 + s_i^2/(\eta_1 s_i^2 + \eta_2)}\right) V^\mathsf{T}, \tag{38}
$$

see Sec. A for the proof. Hence both $W^{-1}v$ and $Y^{-1}u$ reduce to inexpensive diagonal scalings in the SVD basis, avoiding the calculation of any explicit matrix inversion or square-root. The algorithm of MAS for inverse problem is provided in Algorithm 1.

As $\eta_1 < 0$, $W$ could be non-invertible. However, $W = U\mathrm{diag}(\eta_1 s_1^2, \cdots, \eta_1 s_r^2, \eta_2, \cdots, \eta_2) U^T$. Hence $W$ is invertible if $\eta_1 s_i^2 + \eta_2 \neq 0$ for every $i$. Even when $\eta_2 = 0$ and $\eta_1 < 0$ make $W$ singular, the update $W^\dagger y$ uses the Moore-Penrose pseudo-inverse $W^\dagger$, which is always well-defined. The pseudo-inverse acts like an ordinary inverse on the range of $H$ and leaves the null-space untouched, so the sampler remains stable. Empirically, small negative values ($-0.5 < \eta_1 < 0$) often give the visual boost without instability, as demonstrated in the ablation studies in Sec. 5

**Why a negative $\eta_1$ value leads to improvements**? As $\eta_2 = 0$, $x_0^*$ in Eq. (7) can be rewritten as:

$$
x_0^* = m_{0|t} + \frac{1}{1 + \eta_1} H^\dagger (y - H m_{0|t}) = m_{0|t} + \frac{1}{1 + \eta_1} (x_0^{proj} - m_{0|t}) \tag{39}
$$

where

$$
x_0^{proj} := m_{0|t} + H^\dagger y - H^\dagger H m_{0|t} \tag{40}
$$

is the the orthogonal projection of $m_{0|t}$ onto the affine constraint set $\{x : Hx = y\}$. Thus, the update direction $x_0^{proj} - m_{0|t}$ plays the role of a guidance direction, and the scalar $1/(1 + \eta_1)$ acts as guidance strength. When $\eta_1 < 0$, we have $1/(1 + \eta_1) > 1$, i.e., a step larger than the projection step. Such over-guidance (guidance scale > 1) is well-documented in diffusion literature: in particular, Ho & Salimans (2022) and Nichol et al. (2021) all show that over-guidance (scale > 1) improves perceptual fidelity and conditioning strength, at the cost of reduced diversity. Our use of $\eta_1 < 0$ mirrors this phenomenon: a stronger measurement-consistent pull improves reconstruction fidelity under model mismatch, despite departing from the strict probabilistic interpretation.

### B.3 ADDRESSING UNKNOWN NOISE AND NON-DIFFERENTIABLE MEASUREMENTS

As the measurement noise $\epsilon_y$ is non-Gaussian or unknown, it's difficult to ensure the total noise introduced in $x_{t-\Delta t}$ is $\mathcal{N}(0, c_t^2\mathbb{I})$. In this case, we calculate $x_0^*$ using Eq. (7) and update $x_{t-\Delta t}$ using Eq. (2). Then $x_0^*$ is:

$$x_0^* = m_{0|t} + [(Y^{-1} - \mathbb{I})m_{0|t} + H^\mathsf{T}W^{-1}y] \tag{41}$$

$$\tag{42}$$

$$= \underbrace{m_{0|t} + (Y^{-1} - \mathbb{I})m_{0|t} + Y^{-1}H^TW^{-1}y^{\text{clean}}}_{:= \tilde{x}_0^{\text{clean}}} + Y^{-1}H^\mathsf{T}W^{-1}\epsilon_y \tag{43}$$

where $Y^{-1}H^TW^{-1}\epsilon_y$ is the extra noise introduced into $x_0^*$ and will be further introduced into $x_{t-\Delta t}$:

$$x_{t-\Delta t} = a_t x_0^* + b_t x_t + \epsilon_{\text{new}},$$
$$= a_t \tilde{x}_0^{\text{clean}} + b_t x_t + \underbrace{a_t Y^{-1}H^\mathsf{T}W^{-1}\epsilon_y}_{:= \epsilon_{\text{intro}}} + \epsilon_{\text{new}}, \tag{44}$$

where $\epsilon_{\text{new}}$ the noise added by diffusion process, which should be specifically designed to ensure $x_{t-\Delta t}$ is sampled from correct distribution as in Eq. (2), i.e., the total noise $\epsilon_{\text{intro}} + \epsilon_{\text{new}} \sim \mathcal{N}(0, c_t^2\mathbb{I})$. However, as $\epsilon_y$ is unknown noise, we have no information about the introduced noise $\epsilon_{\text{intro}}$. To solve this problem, we made the following principles: (i) despite that fact that we cannot guarantee $\epsilon_{\text{intro}} + \epsilon_{\text{new}} \sim \mathcal{N}(0, c_t^2\mathbb{I})$, we still hope $\epsilon_{\text{intro}} + \epsilon_{\text{new}}$ is as close to $\mathcal{N}(0, c_t^2\mathbb{I})$ as possible; (ii) small $\eta_1$ and $\eta_2$ are helpful to maximize the alignment to measurement $y$. Notably,

$$\epsilon_{\text{intro}} = a_t Y^{-1}H^\mathsf{T}W^{-1}\epsilon_y \tag{45}$$

$$= a_t V \text{diag}\left(\frac{s_i}{(\eta_1 + 1)s_i^2 + \eta_2}\right) U^T \epsilon_y \tag{46}$$

$\eta_1$ and $\eta_2$ are two variables that control the noise level of $\epsilon_{\text{intro}}$. In the implementation, we still sample $\epsilon_{\text{new}}$ from Gaussian distribution $\mathcal{N}(0, c_t^2\mathbb{I})$. Then the problem becomes how to select $\eta_1$ and $\eta_2$ to meet the above 2 principles. For common image restoration tasks like SR, Deblurring, inpainting, Colorization, The maximum eigenvalue value $s_{\max} = \max\{s_i\} <= 1$. Therefore, adjusting $\eta_2$ is more likely to reduce the strength of $\epsilon_{\text{intro}}$.

**Principle (i): Control the deviation from Gaussian reverse noise.** Let

$$A_t = \frac{a_t}{c_t} V \text{diag}\left(\frac{s_i}{(\eta_1 + 1)s_i^2 + \eta_2(t)}\right) U^\mathsf{T}, \qquad \frac{\epsilon_{\text{intro}}}{c_t} = A_t \epsilon_y.$$

To make the total noise

$$\frac{\epsilon_{\text{intro}}}{c_t} + \frac{\epsilon_{\text{new}}}{c_t}$$

as close as possible to $\mathcal{N}(0, \mathbb{I})$, we require the operator norm of $A_t$ to be uniformly bounded:

$$\|A_t\|_2 = \max_i \left|\frac{a_t}{c_t}\frac{s_i}{(\eta_1 + 1)s_i^2 + \eta_2(t)}\right| \leq C. \tag{P1}$$

Since $s_i \leq s_{\max}$, this condition is satisfied if $\eta_2(t) \geq k\,a_t/c_t$, which implies

$$\|A_t\|_2 \leq \frac{s_{\max}}{k}.$$

**Principle (ii): Preserve measurement alignment.**    To preserve measurement alignment, we require $\eta_1$ and $\eta_2$ as small as possible, Considering $\eta_2(t) \geq ka_t/c_t$, the solution to meet Principle (ii) is exactly $\eta_2(t) = ka_t/c_t$, where $k$ is a constant that limit $A_t$ to be uniformly bounded.

## C    LIMITATIONS

While MAS can, in principle, be generalized to nonlinear inverse problems, explicitly formulating the likelihood term $p(y \mid x_\epsilon)$ becomes challenging. Developing effective sampling techniques under this setting is a promising direction for future research.

## D    IMPACT STATEMENT

Our method can improve image restoration under challenging noise and degradation conditions, which may benefit applications in medical imaging, scientific visualization, cultural heritage preservation, and general photography. However, it is important to note that as with many generative and restoration models, our method could be misused for malicious image manipulation.

## E    PYTORCH-LIKE CODE IMPLEMENTATION

Here we provide a basic PyTorch-Like implementation of the calculation of $x_0^*$ in Eq. (7), shown in Listing 1.

**Listing 1** PyTorch-like implementation of the calculation of $x_0^*$ in Eq. (7).

```
@torch.no_grad()
def mas(
    H, x0_hat, y,
    eta_1=-0.2, eta_2=0.0
):
    bs, _, H_img, W_img = x0_hat.shape
    x0_hat = x0_hat.view(bs,  -1)
    y        = y.view( bs, -1)                      # measurement dim m
    ut_y       = H.Ut(y)                            # (bs, m)
    singulars  = H.singulars()                      # (m,)
    nz         = singulars > 0                      # boolean mask
    scale1     = 1.0 / (singulars[nz] ** 2 * eta_1 + eta_2)
    ut_y[:, nz] = ut_y[:, nz] * scale1              # broadcasting OK
    u_y        = H.U(ut_y)                          # (bs, m)
    rhs        = x0_hat + H.Ht(u_y)                 # (bs, d)

    vt_rhs     = H.Vt(rhs)                          # (bs, d)
    scale2     = 1.0 / (1.0 + singulars[nz]**2 / (singulars[nz]**2 *
    ↪  eta_1 + eta_2))
    vt_rhs[:, nz] = vt_rhs[:, nz] * scale2
    x0_pm     = H.V(vt_rhs)                         # (bs, d)
    x0_pm = x0_pm.view(bs, 3, H_img, W_img)
    return x0_pm
```

## F    EXPERIMENTAL DETAILS

### F.1    DETAILS OF THE DEGRADATION OPERATORS

**Super-resolution.** We use the downsampler with bicubic kernel as the forward operator.

Table 5: Sampling time (Sec) per image of MAS on deblurring and super-resolution with FFHQ 256, evaluated using a single NVIDIA A6000 48G GPU. We set NFE=20 and batch size = 20 for all of the methods.

| Method | MAS | ΠGDM | DDNM | DDRM | RED-Diff |
|--------|-----|------|------|------|----------|
| Deblurring | 0.128 | 0.278 | 0.127 | 0.127 | 0.119 |
| SR ($8\times$) | 0.131 | 0.282 | 0.131 | 0.131 | 0.125 |

Table 6: Ablation study of $k$ for solving JPEG restoration and Quantization.

| Method | k | JPEG Restoration (QF = 5) | | | | JPEG Restoration (QF = 2) | | | | Quantization (number of bits = 2) | | | |
|--------|---|------|------|------|------|------|------|------|------|------|------|------|------|
| | | PSNR↑ | SSIM↑ | LPIPS↓ | FID↓ | PSNR↑ | SSIM↑ | LPIPS↓ | FID↓ | PSNR↑ | SSIM↑ | LPIPS↓ | FID↓ |
| ΠGDM | - | 25.78 | 0.750 | 0.241 | 89.82 | 22.92 | 0.653 | 0.314 | 112.27 | 29.98 | 0.823 | 0.185 | 124.57 |
| MAS (ours) | 0.5 | 26.00 | 0.778 | 0.317 | 122.56 | 22.64 | 0.653 | 0.485 | 264.82 | 28.97 | 0.837 | 0.196 | 69.62 |
| | 1.0 | 26.30 | 0.787 | 0.281 | 101.27 | 23.29 | 0.698 | 0.409 | 164.54 | 28.44 | 0.826 | 0.196 | 75.09 |
| | 2.0 | 25.97 | 0.774 | 0.273 | 103.09 | 23.75 | 0.722 | 0.351 | 119.86 | 27.22 | 0.798 | 0.220 | 89.94 |
| | 3.0 | 25.41 | 0.758 | 0.281 | 106.71 | 23.72 | 0.722 | 0.335 | 114.81 | 26.23 | 0.776 | 0.243 | 96.86 |

**Deblurring.** For deblurring experiments, We use uniform blur kernel to to implement blurring operation.

**Inpaint (Random).** Random Inpainting uses a generated random mask where each pixel has a 70% chance of being masked, following the settings in (Song et al., 2023a).

**Inpaint (box).** We use a fixed square mask of size $128 \times 128$ pixels placed at the center of the image.

**Colorization.** We simulate grayscale degradation by applying a fixed linear transformation to each pixel using the matrix $[\frac{1}{3}, \frac{1}{3}, \frac{1}{3}]$, replacing each RGB pixel with its average intensity.

### F.2 DETAILS OF THE BASELINE MODELS

**Sampler**. Most experiments on diffusion models leverage DDIM (Song et al., 2020a) sampling.

**DDRM** (Kawar et al., 2022a). $\eta_B = 1.0$, $\eta = 0.85$ with DDIM sampler, as advised in the original paper.

**ΠGDM** (Song et al., 2023b). $\eta = 1.0$, with DDIM sampler, as advised in the original paper.

**Reddiff** (Mardani et al., 2023). $\lambda = 0.25$, with DDIM sampler, as advised in the original paper.

**DDNM** (Wang et al., 2022). $\eta = 0.85$, with DDIM sampler, as advised in the original paper.

**DAPS** (Zhang et al., 2024). $\tau = 0.01$, with EDM sampler, as advised in the original paper.

## G ADDITIONAL RESULTS

### G.1 COMPUTATIONAL TIME

The computational time of MAS on solving inverse problems is shown in Table 5. Our model achieves similar efficiency to DDNM and DDRM, demonstrating that MAS introduces minimal overhead while maintaining competitive runtime performance.

### G.2 ABLATION STUDY OF $k$

We perform an ablation study of the parameter $k$ for JPEG restoration and quantization (see Table 6). The results indicate that our method is robust to the choice of $k$, and a single setting (e.g., $k = 1$) provides consistently strong performance across all three degradations. Importantly, in contrast to ΠGDM, MAS operates without requiring access to the forward operator.

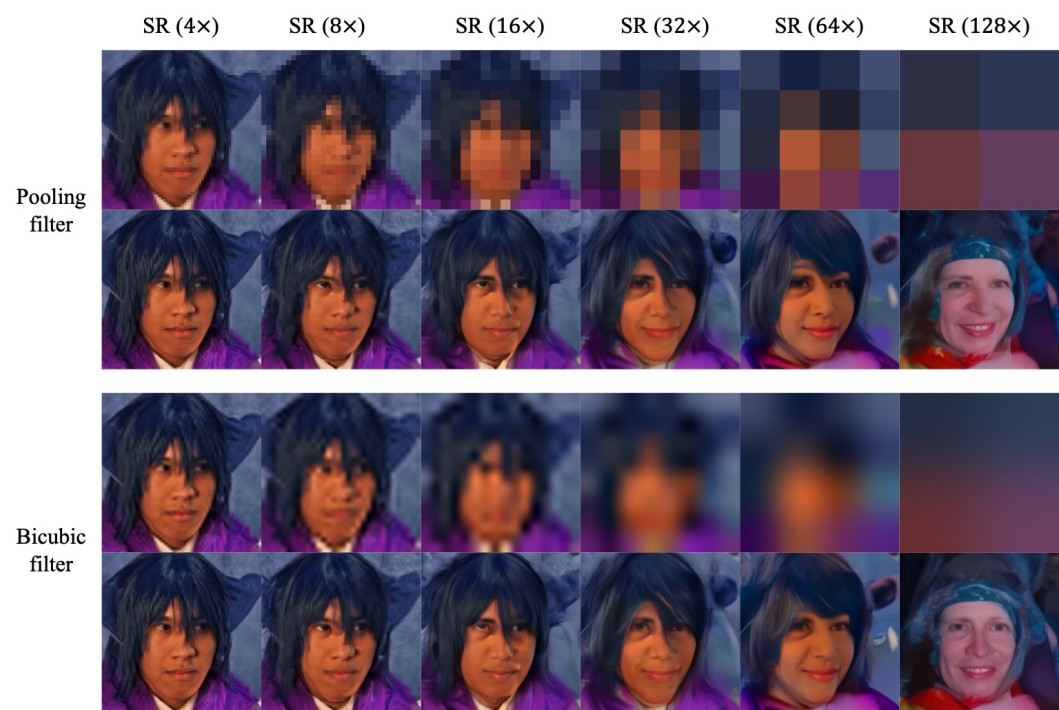

Figure 6: Super-resolution restoration over various strength of degradation. We set $\eta_1 = -0.4$ and $\eta_2 = 0$ for all tasks. For sampling process, we set $\eta = 0.6$.

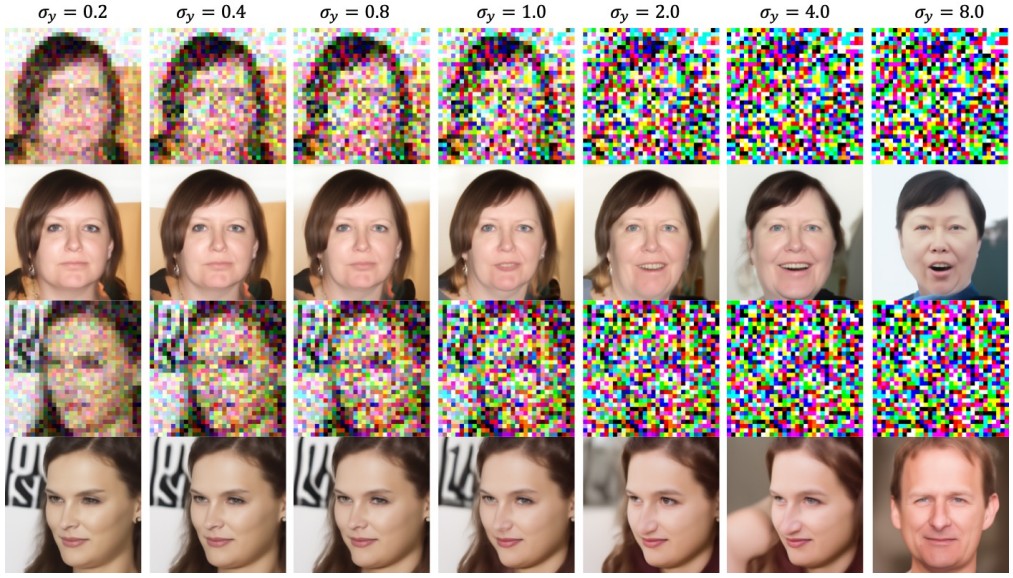

Figure 7: MAS for solving super-resolution ($8\times$) with various strength of Gaussian noise.

### G.3 ADDING GAUSSIAN NOISE

We present the results of solving $8\times$ super-resolution under varying levels of Gaussian noise in Fig. 7. The visualizations demonstrate that MAS maintains strong restoration performance, even under high noise conditions.

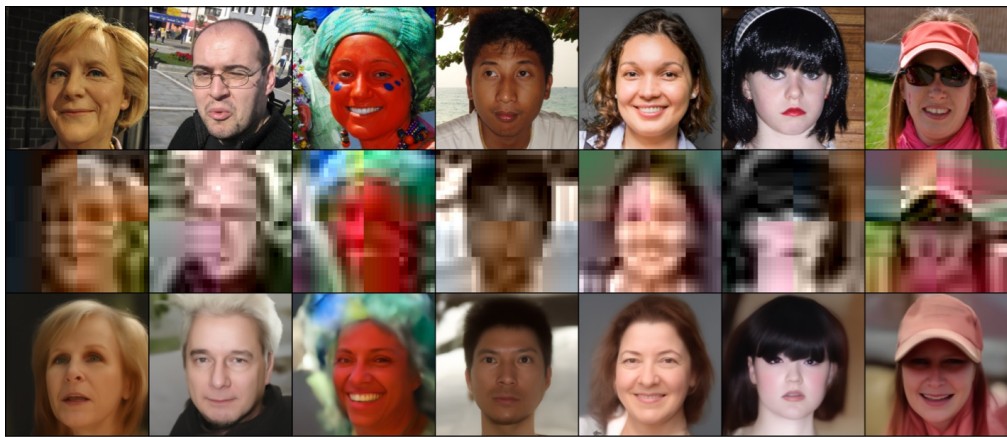

(a) SR(8×) +JPEG

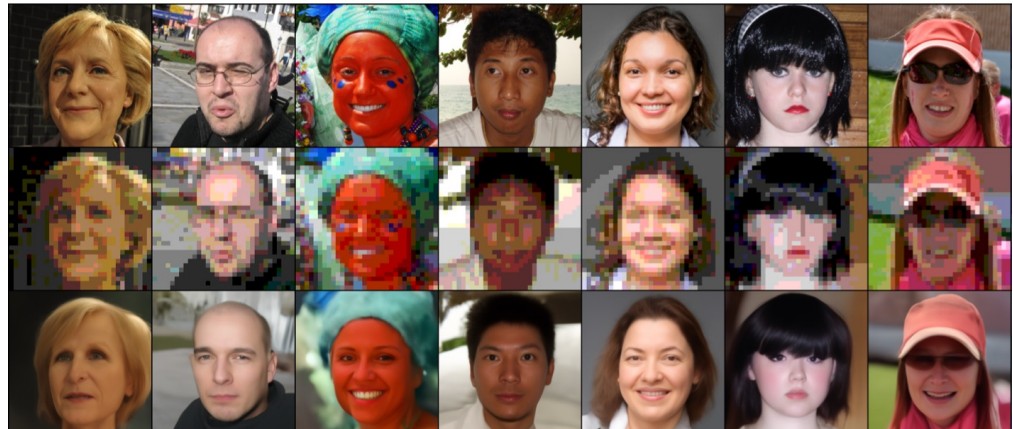

(b) SR (8×) + Quantization

Figure 8: Additional visualization of super-resolution with unknown noise.

### G.4 ADDING NON-DIFFERENTIABLE DEGRAGATION

We present the results of solving $8\times$ super-resolution with non-differentiable degradations, including JPEG compression and quantization, in Fig. 8.

## H LICENSES

**FFHQ Dataset.** We use the Flickr-Faces-HQ (FFHQ) dataset released by NVIDIA under the Creative Commons BY-NC-SA 4.0 license. The dataset is intended for non-commercial research purposes only. More details are available at: `https://github.com/NVlabs/ffhq-dataset`.

**ImageNet Dataset.** The ImageNet dataset is used under the terms of its academic research license. Access requires agreement to ImageNet's data use policy, and redistribution is not permitted. More information is available at: `https://image-net.org/download`.

