# OpenReview forum: "Measurement-Aligned Sampling for Inverse Problem"
_ICLR.cc/2026/Conference — Submitted to ICLR 2026_

### Official Review · Reviewer_pz1H · 2025-10-29

**Soundness:** 3
**Presentation:** 2
**Contribution:** 3
**Rating:** 6
**Confidence:** 4

**Summary:**

The paper generalizes diffusion-based inverse problem solvers into a latent optimization framework by introducing a weighted matrix that balances measurement fidelity and diffusion prior regularization through two hyperparameters. Through an analysis of the solver’s behavior with respect to each hyperparameter, the paper derives appropriate design choices and demonstrates robustness across various types of measurement noise, including Gaussian and non-Gaussian noise, as well as for non-differentiable forward operators.

**Strengths:**

- The framework of optimization problem with weighted matrix that reflects the geometry of the forward operator generalize existing method and eventually provide a better choice that improves the performance.
- The proposed method extends the ability of solving inverse problem with diffusion models to unknown measurement noise.
- The paper provides extensive comparison on various linear inverse problems and baselines.

**Weaknesses:**

- The writing can be improved. Especially, it is quite hard to figure out the most important factor among a lot of parameters such as $\eta_1, \eta_2, c_t, r_t, \sigma_y, \sigma_\epsilon$.
- Discussion on negative $\eta_1$ does not explain the reason why "overshoot" is better than other cases. Appendix B.2 provides a discussion about why "overshoot" does not cause a crucial problem with non-invertible W.

**Questions:**

- From the ablation study result in Figure 4, the performance is gradually better if we use more negative $\eta_1$. If we use smaller value for $\eta_1$ (e.g. $\eta_1=-1.0$), will the performance be continuously better?
- Also, in figure 4, the performance is the best when $\eta_2=0$, which is DDNM according to Remark 1. Is $\eta_2 = \sigma_r^2/r_t^2$ after the proposition 3.1 is better then this setting?

---

> ### Author Response · Authors · 2025-11-18
>
> ## Response to Weakness 1
>
> > The writing can be improved. Especially, it is quite hard to figure out the most important factor among a lot of parameters such as $\eta_1$, $\eta_2$, $c_t$, $r_t$, $\sigma_y$, $\sigma_{\epsilon}$.
>
> Thank you for pointing out that the roles of the different parameters were not sufficiently clear. In the revised manuscript, we have substantially clarified the writing around these parameters and explicitly highlighted which ones are user-facing hyperparameters:
>
> **Emphasizing $\eta_1$ and $\eta_2$ as the main hyperparameters**. In Sec. 3.1, we now clearly state that $\eta_1$ and $\eta_2$ are the primary hyperparameters of MAS.
>
> **Explaining that $r_t$, $\sigma_{\epsilon}$ are not additional hyperparameters**. In Sec. 3.2, $r_t$ and $\sigma_{\epsilon}$ are not additional hyperparameters to be tuned in practice; rather, they serve only to motivate the definitions of $\eta_1$ and $\eta_2$, which are actual user-facing hyperparameters.
>
> **The role of $c_t, \sigma_y$**. $\sigma_y$ is the noise level, and $c_t$ is part of the standard diffusion noise schedule, analogous to prior work such as DDNM+. They are not new MAS-specific hyperparameters.
>
> ## Response to Weakness 2
>
> > Discussion on negative $\eta_1$ does not explain the reason why "overshoot" is better than other cases. Appendix B.2 provides a discussion about why "overshoot" does not cause a crucial problem with non-invertible W.
>
> We now added justification on why a negative $\eta_1$ value leads to improvements in Appendix B.2. As $\eta_2=0$, $x_0^*$ can be rewritten as:
>
> $$    x_0^* = m_{0 \mid t} + \frac{1}{1 + \eta_1} H^{\dagger} (y - Hm_{0 \mid t})= m_{0 \mid t} + \frac{1}{1 + \eta_1} (x_0^{proj} - m_{0 \mid t}),$$
>
> where
>
> $$x_0^{proj}:= m_{0 \mid t} + H^{\dagger} y - H^{\dagger}H m_{0 \mid t} $$
>
> is the the orthogonal projection of $m_{0 \mid t}$ onto the affine constraint set $\{ x: Hx=y \}$. Thus, the update direction $x_0^{proj}-m_{0 \mid t}$ plays the role of a guidance direction, and the scalar $1/(1 + \eta_1)$ acts as guidance strength. When $\eta_1 < 0$, we have $1 / (1 + \eta_1) > 1$, i.e., a step larger than the projection step. Such over-guidance (guidance scale > 1) is well-documented in diffusion literature: in particular,[R1] and [R2] all show that over-guidance (scale $> 1$) improves perceptual fidelity and conditioning strength, at the cost of reduced diversity. Our use of $\eta_1 < 0$ mirrors this phenomenon: a stronger measurement-consistent pull improves reconstruction fidelity under model mismatch, despite departing from the strict probabilistic interpretation.
>
> ## Response to Question 1
>
> > From the ablation study result in Figure 4, the performance is gradually better if we use more negative $\eta_1$. If we use smaller value for $\eta_1$ (e.g., $\eta_1=-1.0$), will the performance be continuously better?
>
> Theoretically, as $\eta_2=0$, $x_0^*$ can be rewritten as:
>
> $$    x_0^* = m_{0 \mid t} + \frac{1}{1 + \eta_1} H^{\dagger} (y - Hm_{0 \mid t})= m_{0 \mid t} + \frac{1}{1 + \eta_1} (x_0^{proj} - m_{0 \mid t}),$$
>
> Therefore, we require $\eta_1>-1$. Since $\frac{1}{1 + \eta_1}$ controls the ‘strength’ of guidance, $\eta_1$ can not be too small.
>
> Empirically, we revised the ablation study in Fig 4 to extend the range of $\eta_1$. From the results we observed performance decreased as we set $\eta_1=-0.55$ in the Super-resolution and $\eta_1=-0.5$ in the Deblurring task.
>
> ## Response to Question 2
>
> > Also, in figure 4, the performance is the best when $\eta_2=0$, which is DDNM according to Remark 1. Is $\eta_2=\sigma_r^2/r_t^2$ after the proposition 3.1 is better then this setting?
>
> There’s no difference between these two settings, because $r_t$ is an unknown. For every choice of $r_t$, we could find a related choice of $\eta_2(t)$. Here we use $\eta_2=\sigma_y^2/r_t^2$ to provide a probabilistic explanation for $\eta_2$.

---

### Official Review · Reviewer_QmhR · 2025-11-01

**Soundness:** 3
**Presentation:** 3
**Contribution:** 2
**Rating:** 4
**Confidence:** 4

**Summary:**

The paper proposes a new framework called Measurement-Aligned Sampling (MAS), a unified method for solving linear inverse problems using diffusion models that generalizes and improves upon previous approaches like DDNM and TMPD. This framework is designed to better balance prior knowledge with measurement data, and demonstrates the robustness to handle unknown, non-Gaussian, and non-differentiable noise without requiring prior knowledge of the noise structure. Extensive experiments validate the effectiveness of MAS and show that MAS consistently outperforms other state-of-the-art methods across various tasks.

**Strengths:**

1. The paper introduces effective techniques for handling both Gaussian noise and unknown noise sources.

2. The proposed method enhances the robustness of diffusion models in inverse problems.

**Weaknesses:**

1. This paper proposes an adaptive parameter scheme with $\eta_2 = ka_t / c_t$ to generalize approaches for linear inverse problems with diffusion models. This scheme seems to be heuristic. According to Appendix B.3, this scheme is obtained based on informal principles such as "hoping $\epsilon_{intro} + \epsilon_{new}$ is as close to $\mathcal{N}(0, c_t^2 \mathbb{I})$ as possible". There is no rigorous theoretical justification on the reason why $\eta_2$ must be proportional to $a_t/c_t$.

2. The reason why a negative $\eta_1$ value leads to improvements is not well explained. It is not clear whether this is related to compensating for inherent biases in the pre-trained diffusion model or there is any geometric interpretation in the optimization objective (Equation (6)) that differs from the probabilistic model (e.g., as a super-linear interpolation between $m_{0|t}$ and the DDNM solution).

3. Review on related works is limited. The paper claims to generalize approaches for linear inverse problems with diffusion models, but neglects related works with similar claims such as [R1]-[R3]. Discussion on the difference from these works and necessary comparison are required to clarify the contributions claimed in this paper.

4. The caption of Table 4 explicitly states that the method requires manually setting different k values for different degradation tasks (e.g., k=1.0 for JPEG QF=5 and k=3.0 for QF=2). It appears to require the user to have prior knowledge of the degradation type to select an appropriate k. This undermines the ability to handle "unknown" noise as claimed in the paper.

[R1] Yismaw N, Kamilov U S, Asif M S. Gaussian is all you need: A unified framework for solving inverse problems via diffusion posterior sampling. IEEE Transactions on Computational Imaging, 2025.

[R2] Peng X, Zheng Z, Dai W, Xiao N, Li C, Zou J, Xiong H. Improving Diffusion Models for Inverse Problems Using Optimal Posterior Covariance. International Conference on Machine Learning. 2024: 40347-40370.

[R3] Fei B, Lyu Z, Pan L, Zhang J, Yang W, Luo T, Zhang B, Dai B. Generative Diffusion Prior for Unified Image Restoration and Enhancement. Proceedings of the IEEE/CVF Conference on Computer Vision and Pattern Recognition (CVPR), 2023, pp. 9935-9946.

**Questions:**

Please refer to Weaknesses.

---

> ### Author Response · Authors · 2025-11-18
>
> ## Response to Weakness 1
>
> We provided theoretical justification on the reason why we choose $\eta_2$ to be proportional to $a_t/c_t$, see details in Appendix B3 in the revised paper.
>
> ## Response to Weakness 2
>
>
> We added justification on why a negative $\eta_1$ value leads to improvements in Appendix B.2. As $\eta_2=0$, $x_0^*$ can be rewritten as:
>
> $$    x_0^* = m_{0 \mid t} + \frac{1}{1 + \eta_1} H^{\dagger} (y - Hm_{0 \mid t})= m_{0 \mid t} + \frac{1}{1 + \eta_1} (x_0^{proj} - m_{0 \mid t}),$$
>
> where
>
> $$x_0^{proj}:= m_{0 \mid t} + H^{\dagger} y - H^{\dagger}H m_{0 \mid t} $$
>
> is the the orthogonal projection of $m_{0 \mid t}$ onto the affine constraint set $\{ x: Hx=y \}$. Thus, the update direction $x_0^{proj}-m_{0 \mid t}$ plays the role of a guidance direction, and the scalar $1/(1 + \eta_1)$ acts as guidance strength. When $\eta_1 < 0$, we have $1 / (1 + \eta_1) > 1$, i.e., a step larger than the projection step. Such over-guidance (guidance scale > 1) is well-documented in diffusion literature: in particular,[R5] and [R6] all show that over-guidance (scale $> 1$) improves perceptual fidelity and conditioning strength, at the cost of reduced diversity. Our use of $\eta_1 < 0$ mirrors this phenomenon: a stronger measurement-consistent pull improves reconstruction fidelity under model mismatch, despite departing from the strict probabilistic interpretation.
>
> ## Response to Weakness 3
>
> First, [R1] and [R2] did not break out of the **Tweedie Moment Projected Diffusion (TMPD)** framework [R4], where $p(x_0 \mid x_t)$ was assumed as a Gaussian: $p(x_0 \mid x_t)\approx \mathcal{N}(m_{0\mid t}, C_{0\mid t})$. For linear inverse problem,
> $$p(y \mid x_t) = \mathcal{N} (y; H m_{0 \mid t}, HC_{0 \mid t} H^T + \sigma_y^2 \mathbb{I}).$$
>
> Specifically, [R1] derives closed-form expressions for the approximated $p(x_0 \mid x_t)$ in Eqs. (10)-(11), We note that these formulas already appeared in **TMPD** (See Eqs. (10)-(11) in the original TMPD paper). [R2] claimed that it unifies the existing approaches with  isotropic Gaussian approximations $p(x_0 \mid x_t) = \mathcal{N}(x_0 \mid D_t(x_0), r_t^2 \mathbb{I})$ to the intractable denoising posterior $p_t(x_0 \mid x_t)$ with different $r_t$. In contrast, **TMPD** (and thus MAS) uses the **non-isotropic** Tweedie posterior: $\mathcal{N} (m_{0 \mid t}, C_{0 \mid t})$. Therefore, TMPD provides a more generalized framework than [R2]. In our work, we have detailed comparison between our work and TMPD, see Proposition 3.1 and Remark 3.
>
> Second, [R3] propose a heuristic approximation of $p(y \mid x_t)$:
>
> $$p(y \mid x_t) = \frac{1}{Z} \exp (- [s \mathcal{L} (D(x_t), y) + \lambda \mathcal{Q} (x_t)])$$
>
> They do unify current works by allowing different choices of $\mathcal{L}$ and $\mathcal{Q}$. But it is not specifically designed for linear inverse problems with known forward operator $H$. In contrast, TMPD gives a more accurate approximation for linear inverse problems.
>
> We have revised Sec. 2 to explicitly position TMPD/MAS relative to [R1–R3]. The updated text now reads:
> - "Since $\nabla_{x_t} \log p(y \mid x_t)$ is generally intractable, various approaches have been developed to approximate it, such as heuristic approximation [R3], For linear inverse problems, where the forward model is given by: $y = Hx_0 + \epsilon$, $\epsilon \sim \mathcal{N}(0, \sigma_y^2 \mathbb{I})$. Tweedie Moment Projected Diffusion (TMPD) [R4] provides a more accurate approximation."
>
> - "Since calculating the gradient with respect to $m_{0 \mid t}$ is time-consuming, [R1] and [R2] try to find the optimal isotropic approximation of $C_{0 \mid t}$, i.e., $C_{0 \mid t} \approx r_t^2 \mathbb{I}$."
>
> ## Response to Weakness 4
>
> **k is not a sensitive parameter**. We have added an ablation study covering three representative tasks—JPEG restoration (QF=5), JPEG restoration (QF=2), and quantization (2-bit)—in Appendix G.2 and Table 6. In all cases, we sweep k from a set {0.5, 1.0, 2.0, 3.0}. The results consistently show that the method performs robustly across a wide range of $k$. For instance, in the JPEG restoration (QF=5), the PSNRs obtained with different $k$ values are:  26.00, 26.30, 25.97, 25.41 respectively— all competitive with the baseline PSNR of 25.78.
>
>
> **The optimal k depends on the strength of the noise, and we provide theoretical guidance**. In the Appendix B3, we added the theoretical justification on  why we set $\eta_2=ka_t/c_t$, how $k$ controls the amplification of unknown noise in $x_t$. This analysis provides a principled guideline: $k$ balances two competing effects—maintaining measurement consistency and limiting noise injection.
> As noise severity increases, a slightly larger $k$ helps suppress amplified noise. This trend matches our empirical observations. For instance, the optimal $k$ for JPEG restoration at QF=5 is 1.0, whereas for the more challenging QF=2 setting—with substantially greater information loss—the optimal $k$ increases to 2.0.

---

### Official Review · Reviewer_PMYZ · 2025-11-03

**Soundness:** 3
**Presentation:** 3
**Contribution:** 2
**Rating:** 6
**Confidence:** 2

**Summary:**

This paper introduces Measurement-Aligned Sampling (MAS), a framework for solving linear inverse problems using diffusion models. The method optimizes a weighted objective balancing prior and measurement fidelity through parameters η₁ and η₂, with closed-form solutions via SVD. MAS generalizes DDNM and TMPD, introduces an overshooting technique, and proposes adaptive parameterization for unknown noise. Experiments show state-of-the-art performance on super-resolution, inpainting, deblurring, JPEG restoration, and quantization tasks.

**Strengths:**

1.  The paper provides both probabilistic (Bayesian linear regression) and optimization perspectives for a single weighted objective balancing prior and measurement fidelity, with efficient closed-form solutions via SVD decomposition.

2. MAS outperforms baselines in several tasks, while maintaining efficiency comparable to DDNM.

3. The adaptive parameterization enables effective restoration on real-world degradations like JPEG and quantization without requiring exact noise specifications, though limited to problems approximable as linear with unknown noise.

**Weaknesses:**

1. The k parameter in lacks principled selection criteria. Table 4 shows different values (k = 1.0, 3.0, 0.5) without rationale, which might require expensive grid searches for new tasks. This contradicts the claimed relatively low computational cost.

2.  The probabilistic foundation defines η₁ ≥ 0, yet optimal results (Figure 4) use negative η₁ = -0.45. While numerical stability is addressed, the paper lacks justification for why violating this constraint improves performance.

3.  While Table 3 provides quantitative evaluation for salt-and-pepper and periodic noise, Poisson noise appears only qualitatively in Figure 1. This leaves unclear whether the method quantitatively generalizes across diverse unknown noise types meaningfully.

**Questions:**

1. Can the authors provide principled guidelines for selecting k based on observable degradation characteristics? How sensitive is the method's performance to k values?

2. Can the authors provide an intuition for why negative η₁ empirically improves results despite violating the probabilistic interpretation?

---

> ### Author Response · Authors · 2025-11-18
>
> W1 and Q1
>
>
> > The k parameter lacks principled selection criteria. Table 4 shows different values (k = 1.0, 3.0, 0.5) without rationale, which might require expensive grid searches for new tasks. This contradicts the claimed relatively low computational cost.
>
> > Can the authors provide principled guidelines for selecting k based on observable degradation characteristics? How sensitive is the method's performance to k values?
>
> Thank you for the thoughtful comment regarding the selection of the parameter $k$. Below we clarify both the empirical behavior and the theoretical motivation.
>
> **k is not a sensitive parameter**. We have added an ablation study covering three representative tasks—JPEG restoration (QF=5), JPEG restoration (QF=2), and quantization (2-bit)—in Appendix G.2 and Table 6. In all cases, we sweep k from a set {0.5, 1.0, 2.0, 3.0}. The results consistently show that the method performs robustly across a wide range of $k$. For instance, in the JPEG restoration (QF=5), the PSNRs obtained with different $k$ values are: 26.00, 26.30, 25.97, 25.41 respectively— all competitive with the baseline PSNR of 25.78. Importantly, the baseline method requires access to the forward operator, while our approach does not. This indicates that even without precise operator knowledge, our method maintains strong performance without the need for expensive or fine-grained tuning.
>
> **The optimal k depends on the strength of the noise, and we provide theoretical guidance**. In the Appendix B3, we added the theoretical justification on  why we set $\eta_2=ka_t/c_t$, how $k$ controls the amplification of unknown noise in $x_t$. This analysis provides a principled guideline: $k$ balances two competing effects—maintaining measurement consistency and limiting noise injection.
> As noise severity increases, a slightly larger $k$ helps suppress amplified noise. This trend matches our empirical observations. For instance, the optimal $k$ for JPEG restoration at QF=5 is 1.0, whereas for the more challenging QF=2 setting—with substantially greater information loss—the optimal $k$ increases to 2.0.
>
>
> Overall, both theory and experiments demonstrate that (i) performance is robust to a broad range of $k$, and (ii) when needed, the degradation severity provides a clear and principled signal for adjusting $k$.
>
>
> W2 and Q2
> > The probabilistic foundation defines $\eta_1 \geq 0$, yet optimal results (Figure 4) use negative $\eta_1=-0.45$. While numerical stability is addressed, the paper lacks justification for why violating this constraint improves performance.
> > Can the authors provide an intuition for why negative $eta_1$ empirically improves results despite violating the probabilistic interpretation?
>
> We added justification on why a negative $\eta_1$ value leads to improvements in Appendix B.2. As $\eta_2=0$, $x_0^*$ can be rewritten as:
>
> $$    x_0^* = m_{0 \mid t} + \frac{1}{1 + \eta_1} H^{\dagger} (y - Hm_{0 \mid t})= m_{0 \mid t} + \frac{1}{1 + \eta_1} (x_0^{proj} - m_{0 \mid t}),$$
>
> where
>
> $$x_0^{proj}:= m_{0 \mid t} + H^{\dagger} y - H^{\dagger}H m_{0 \mid t} $$
>
> is the orthogonal projection of $m_{0 \mid t}$ onto the affine constraint set $\{ x: Hx=y \}$. Thus, the update direction $x_0^{proj}-m_{0 \mid t}$ plays the role of a guidance direction, and the scalar $1/(1 + \eta_1)$ acts as guidance strength. When $\eta_1 < 0$, we have $1 / (1 + \eta_1) > 1$, i.e., a step larger than the projection step. Such over-guidance (guidance scale > 1) is well-documented in diffusion literature: in particular,[R1] and [R2] all show that over-guidance (scale $> 1$) improves perceptual fidelity and conditioning strength, at the cost of reduced diversity. Our use of $\eta_1 < 0$ mirrors this phenomenon: a stronger measurement-consistent pull improves reconstruction fidelity under model mismatch, despite departing from the strict probabilistic interpretation.
>
>
>
> W3
> > While Table 3 provides quantitative evaluation for salt-and-pepper and periodic noise, Poisson noise appears only qualitatively in Figure 1. This leaves unclear whether the method quantitatively generalizes across diverse unknown noise types meaningfully.
>
> We added quantitative evaluation on solving linear inverse problems with Poisson noise in Table 3. Our MAS still can achieve the best performance across all metrics.
>
>
> [R1]. Ho J, Salimans T. Classifier-free diffusion guidance[J]. arXiv preprint arXiv:2207.12598, 2022.
>
> [R2]. Nichol A, Dhariwal P, Ramesh A, et al. Glide: Towards photorealistic image generation and editing with text-guided diffusion models[J]. arXiv preprint arXiv:2112.10741, 2021.

---

### Author Response · Authors · 2025-12-03
**General response**

We thank the reviewers for their constructive feedback. We are encouraged by the recognition of our method’s soundness (R1, R2, R3) and its state-of-the-art performance across various tasks (R1, R2). We have uploaded a revised manuscript incorporating the suggestions. Below is a summary of the key updates:

**Justification for Negative $\eta_1$ (Appendix B.2)**. We added a geometric interpretation explaining that  acts as "over-guidance" (scale $> 1$). Similar to Classifier-Free Guidance in generation tasks, this effectively pulls the sample stronger towards the measurement subspace, improving fidelity even if it departs from the strict probabilistic path.

**Robustness of Parameter $k$ (Appendix G.2 & Table 6)**. We added an ablation study sweeping $k \in \{0.5, 1.0, 2.0, 3.0\}$. Results show the method is highly robust with minimal performance variance. We also added theoretical analysis (Appendix B.3) confirming that $k$ balances measurement consistency vs. noise amplification, providing a clear rule for adjustment if needed.

**Generalization to Poisson Noise (Table 3)**. To demonstrate generalization beyond Gaussian noise, we added quantitative experiments with Poisson noise. MAS consistently outperforms state-of-the-art baselines on these tasks as well.

**Parameter Clarification (Section 3.1 & 3.2)**. We revised the text to explicitly distinguish between user-facing hyperparameters ($\eta_1, \eta_2$) and internal diffusion variables ($r_t, \sigma_\epsilon$), clarifying that the user does not need to tune a complex set of parameters.

**Related Works (Section 2)**. We expanded the discussion to compare MAS against recent works (e.g., [R1]-[R3]), highlighting how our method unifies and generalizes these approaches.

[R1] Yismaw N, Kamilov U S, Asif M S. Gaussian is all you need: A unified framework for solving inverse problems via diffusion posterior sampling. IEEE Transactions on Computational Imaging, 2025.

[R2] Peng X, Zheng Z, Dai W, Xiao N, Li C, Zou J, Xiong H. Improving Diffusion Models for Inverse Problems Using Optimal Posterior Covariance. International Conference on Machine Learning. 2024: 40347-40370.

[R3] Fei B, Lyu Z, Pan L, Zhang J, Yang W, Luo T, Zhang B, Dai B. Generative Diffusion Prior for Unified Image Restoration and Enhancement. Proceedings of the IEEE/CVF Conference on Computer Vision and Pattern Recognition (CVPR), 2023, pp. 9935-9946.

---

### Meta-Review · Area_Chair_ZpZo · 2026-01-09

**Summary:**

The paper introduces a unified diffusion-based framework for linear inverse problems that balances measurement fidelity and diffusion priors through a weighted objective governed by hyperparameters. This framework integrates prior methods and extends to unknown measurement noise levels and noise modeling such as Gaussian, Poisson, or non-differentiable noise. Experiments are conducted across various tasks including super-resolution, inpainting, deblurring, JPEG restoration, and quantization.

The paper offers both a probabilistic and optimization perspective of a unified weighted objective that balances prior and measurement fidelity, and derives efficient closed-form updates using SVD. The most interesting contribution is to enhance the robustness to both Gaussian and unknown noise sources. The adaptive parameterization enables restoration under real-world degradations such as JPEG and quantization without requiring precise noise specifications, though limited to problems approximable as linear with unknown noise.

Most reviewers raise the same question on the “overshooting” trick used in the paper due to lack of theoretical justification for this key design choice: why using negative values of η₁ improves performance and how it is related to model bias or the optimization geometry. In addition, the proposed adaptive parameterization for unknown noise or non-Gaussian noise and non-differentiable measurements might be heuristic, which is claimed as one of the key contributions distinguished from previous works. More rigorous analysis or justification for this assumption would help to strengthen the paper. Empirical evaluation of robustness to unknown noise is incomplete, as Poisson noise is shown only qualitatively. Finally, there are also concerns about hyper-parameter finetuning and choices, and related-work discussion.

In the rebuttal, the author provides justification and new results to answer the reviewers’ questions, such as ablation study with new parameter choices, and quantitative results for Poisson noises, which improves the paper and answers those questions well. For the justification on the “overshooting” trick negative values of η₁ and the adaptive parameterization may need more discussion. For example, the justification of over-guidance is based on the CFG in text-to-image generation, which is an absolutely different task for solving inverse problems and the evaluation metrics are emphasized from different perspectives. A full discussion from reviewers or another round of review may be helpful to address these remaining concerns. However, none of the reviewers provides feedback and follow-ups responding to the questions.

Overall, I think the paper is on the fence of borderline while leaning to a minor reject due to the remaining concerns and missing discussions as above.

**Reviewer Concerns:**

In the rebuttal, the author provides justification and new results to answer the reviewers’ questions, such as ablation study with new parameter choices, and quantitative results for Poisson noises, which improves the paper and answers those questions well. For the justification on the “overshooting” trick negative values of η₁ and the adaptive parameterization may need more discussion. For example, the justification of over-guidance is based on the CFG in text-to-image generation, which is an absolutely different task for solving inverse problems and the evaluation metrics are emphasized from different perspectives. A full discussion from reviewers or another round of review may be helpful to address these remaining concerns.

**Reviewer Scores:**

Unfortunately, none of the reviewers provides feedback and follow-ups responding to the questions. The original scores from three reviewers are all around borderlines.

---

### Decision · Program_Chairs · 2026-01-26

Reject